# Modifying redox properties and local bonding of Co$_3$O$_4$ by CeO$_2$ enhances oxygen evolution catalysis in acid

Jinzhen Huang[1,2], Hongyuan Sheng [1], R. Dominic Ross[1], Jiecai Han[2], Xianjie Wang[3], Bo Song[2✉] & Song Jin [1✉]

Developing efficient and stable earth-abundant electrocatalysts for acidic oxygen evolution reaction is the bottleneck for water splitting using proton exchange membrane electrolyzers. Here, we show that nanocrystalline CeO$_2$ in a Co$_3$O$_4$/CeO$_2$ nanocomposite can modify the redox properties of Co$_3$O$_4$ and enhances its intrinsic oxygen evolution reaction activity, and combine electrochemical and structural characterizations including kinetic isotope effect, pH- and temperature-dependence, in situ Raman and ex situ X-ray absorption spectroscopy analyses to understand the origin. The local bonding environment of Co$_3$O$_4$ can be modified after the introduction of nanocrystalline CeO$_2$, which allows the Co$^{III}$ species to be easily oxidized into catalytically active Co$^{IV}$ species, bypassing the potential-determining surface reconstruction process. Co$_3$O$_4$/CeO$_2$ displays a comparable stability to Co$_3$O$_4$ thus breaks the activity/stability tradeoff. This work not only establishes an efficient earth-abundant catalysts for acidic oxygen evolution reaction, but also provides strategies for designing more active catalysts for other reactions.

[1] Department of Chemistry, University of Wisconsin–Madison, Madison, WI, USA. [2] Center for Composite Materials and Structures, Harbin Institute of Technology, Harbin, China. [3] School of Physics, Harbin Institute of Technology, Harbin, China. ✉email: songbo@hit.edu.cn; jin@chem.wisc.edu

The fast depletion of fossil fuels and increasing greenhouse effect demand sustainable strategies to produce carbon-neutral fuels using renewable electricity[1]. Electrocatalytic water splitting has been considered a promising approach to generate hydrogen as a clean and renewable energy carrier[2]. Proton exchange membrane (PEM) electrolyzers operated in acidic media have shown great promises for large-scale applications[3–5]. Despite substantial recent advances in the discovery of robust and active earth-abundant electrocatalysts for acidic hydrogen evolution reaction (HER)[1,6–8], the development of high-performance yet cost-effective electrocatalysts for the sluggish four-electron oxygen evolution reaction (OER) is challenging[9–11] especially in acidic media, which contributes to a major energy loss in the overall water splitting process and is a bottleneck for realizing practical PEM electrolyzers[3,12]. Most OER catalysts show inferior activities in acidic media compared to in alkaline media and require higher overpotentials to achieve comparable catalytic current densities. Moreover, the stability issues are more severe in acidic OER, and even noble metal-based catalysts (such as $RuO_2$ and $IrO_2$) experience dissolution and degradation[13,14]. Furthermore, the often observed tradeoff between activity and stability in acidic OER catalysts[13–16] complicates the catalyst design. As a result, there have been very limited choices of earth-abundant OER catalysts that are both active and stable in acidic media[17–20]. Cobalt (Co)-based catalysts such as Ba[Co-POM][17], hetero-N-coordinated Co single atom catalyst[21], $CoFePbO_x$[18], $Co_2TiO_4$[22], and $Co_3O_4$[23–25] are promising for acidic OER; however, the mechanistic details have rarely been studied for these emerging OER catalysts in acidic media.

The active site structures and catalytic mechanisms of cobalt oxide OER catalysts have been primarily investigated in alkaline and neutral media[26–31], little is known about these catalysts in acidic media. The exact configuration of the active sites responsible for the O-O bond formation still remains debatable, but the generation of high-valence-state $Co^{IV}$ is accepted to be involved in the pre-OER redox processes of different types of cobalt oxide OER catalysts since they share the common active sites[26,31,32]. The further oxidation of the neighboring Co redox centers to form dimeric $Co^{IV}Co^{IV}$ takes place at high potentials[33,34], and thus causes a large energy loss to bypass this potential-determining process for the catalytic OER[31]. Besides, these prominent pre-OER redox features also suggest that the $Co^{IV}Co^{IV}$ intermediates are stabilized and could suffer from a slow catalytic turnover process for OER[35,36]. Therefore, a better understanding of the relationships between redox properties and catalytic activity is the key to design more efficient (Co-based) OER catalysts and to enhance catalytic activity by regulating redox properties, which remains elusive and largely underexplored especially in acidic media.

In this work, we enhance the intrinsic catalytic activity of $Co_3O_4$ by introducing nanocrystalline $CeO_2$ to form a heterogeneous $Co_3O_4/CeO_2$ nanocomposite and establish $Co_3O_4/CeO_2$ nanocomposite as an active acidic OER catalyst. $CeO_2$ has been well documented as (co-)catalyst in thermal catalysis due to its excellent redox properties and oxygen storage capacity[37]. Although $CeO_2$ has been introduced into a number of electrocatalyst systems to enhance the overall performance for various electrocatalytic reactions[38] including the alkaline OER[39–41], how it impacts the catalytic activity remains controversial and its contribution to the redox properties of the electrocatalysts has not yet been discussed. Now we show that the introduction of $CeO_2$ (meaning phase-pure $CeO_2$ nanocrystallites are interdispersed among phase-pure $Co_3O_4$ crystallites in the two-component nanocomposite without phase mixing) substantially suppresses the pre-OER redox features of $Co_3O_4$ in acidic media, indicating the destabilization of the dimeric $Co^{IV}Co^{IV}$ intermediate.

In-depth electrochemical characterizations combined with rigorous structural characterizations, including kinetic isotope effect (KIE), pH- and temperature-dependence studies, in situ Raman, and ex situ X-ray absorption spectroscopy (XAS) analyses, reveal that the catalytic enhancement in $Co_3O_4/CeO_2$ is due to the altered electronic structures and local bonding environment in $Co_3O_4$. Chronopotentiometry test together with inductively coupled plasma mass spectrometry (ICP-MS) analysis shows that the more active $Co_3O_4/CeO_2$ exhibits a comparable acidic OER stability to $Co_3O_4$ and a better open circuit stability, thus breaks the activity/stability tradeoff.

## Results and discussion

**Synthesis and structural characterization of $Co_3O_4/CeO_2$ nanocomposites.** $Co_3O_4$ nanostructures and $Co_3O_4/CeO_2$ nanocomposites were synthesized directly on fluorine-doped tin oxide (FTO) electrodes by electrodeposition of the corresponding metal hydroxide precursors (Supplementary Fig. 1) followed by annealing in air (see Methods for details). The prototypical $Co(OH)_2$ precursor displayed the morphology of interconnected nanosheets, while the introduction of Ce precursor led to more aggregations and wrinkles (Supplementary Fig. 2). After annealing in air at 400 °C for 2 h, the resultant $Co_3O_4$ and $Co_3O_4/CeO_2$ samples preserved the nanosheet morphology (Supplementary Fig. 3). High-resolution transmission electron microscopy (HRTEM) further revealed the nanocrystalline domains in both $Co_3O_4$ (Fig. 1a, c and Supplementary Fig. 4a, b) and $Co_3O_4/CeO_2$ (Fig. 1b, d and Supplementary Fig. 4c, d) samples. Because the spinel oxide $Co_3O_4$ and cubic $CeO_2$ structures (Supplementary Fig. 9a) cannot form mixed solutions, phase segregation is expected[42], which is further proved by the powder X-ray diffraction (PXRD) pattern of $Co_3O_4/CeO_2$ (Fig. 1e). Selected area electron diffraction patterns of both samples displayed similar diffraction rings due to the polycrystalline nature (insets of Fig. 1a, b). The inner to outer diffraction rings can be indexed to the (111), (220), (311), (400), (511), (440) planes of $Co_3O_4$ (JCPDS 43-1003), consistent with the PXRD patterns (Fig. 1e) and the spinel oxide crystal structure of $Co_3O_4$ (Fig. 1f)[43]. The introduction of $CeO_2$ decreased the crystallinity of $Co_3O_4$, as the average crystalline domain sizes of $Co_3O_4$ and $Co_3O_4/CeO_2$ estimated from the widths of the (311) diffraction peaks using the Scherrer equation were 13.9 and 9.7 nm, respectively (Supplementary Fig. 5). From the HRTEM images (Fig. 1c, d), the lattice spacings of 0.243 and 0.467 nm were assigned to the (311) and (111) planes of $Co_3O_4$, respectively, and that of 0.312 nm was attributed to the (111) plane of $CeO_2$. Nanoscale crystallites of $CeO_2$ exhibit an average domain size of ~5 nm based on the Scherrer analysis of the PXRD peak (Supplementary Fig. 6) and are evenly dispersed among phase-pure $Co_3O_4$ crystallites with numerous interfacial contact regions. Elemental mappings further confirmed the successful introduction of Ce in $Co_3O_4/CeO_2$ (Fig. 1g). The bulk and surface Ce metal contents in $Co_3O_4/CeO_2$ [defined as $Ce/(Ce + Co) \times 100\%$] were determined as 9.1 and 6.6 atomic percent (at%) using energy-dispersive X-ray spectroscopy (EDS) and X-ray photoelectron spectroscopy (XPS), respectively (Supplementary Table 1).

**Electrocatalytic properties of $Co_3O_4/CeO_2$ nanocomposites.** The substantial differences in the redox properties and acidic OER catalytic performances between the $Co_3O_4$ and $Co_3O_4/CeO_2$ catalysts on FTO electrodes are shown by cyclic voltammetry (CV) recorded in 0.5 M $H_2SO_4$ solution (Fig. 2a). Three sets of pre-OER redox features are observed in $Co_3O_4$ (the corresponding cathodic peaks are denoted as C1, C2, and C3 in the order of increasing potential, see Fig. 2b), which can be ascribed

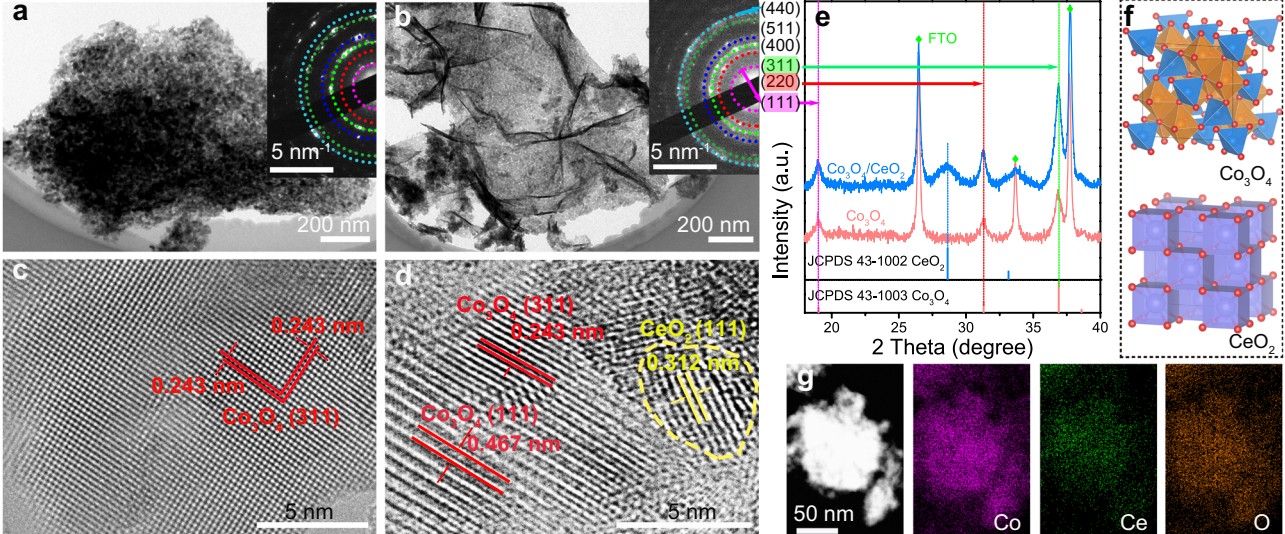

**Fig. 1 Structural characterizations of Co$_3$O$_4$ nanostructures and Co$_3$O$_4$/CeO$_2$ nanocomposites.** TEM images of **a** Co$_3$O$_4$ and **b** Co$_3$O$_4$/CeO$_2$ nanosheets, the insets show the corresponding SAED patterns. HRTEM images of **c** Co$_3$O$_4$ and **d** Co$_3$O$_4$/CeO$_2$ samples. The CeO$_2$ domain is highlighted with a yellow dashed circle. **e** PXRD patterns of the samples on FTO substrates in comparison with the standard PXRD patterns of Co$_3$O$_4$ (JCPDS 43-1003) and CeO$_2$ (JCPDS 43-1002). **f** Crystal structures of Co$_3$O$_4$ and CeO$_2$. **g** Dark-field TEM image and the corresponding elemental mappings of Co, Ce, and O in the Co$_3$O$_4$/CeO$_2$ sample.

to the following equilibria involving dimeric Co redox centers[26,31,33]: Co$^{II}$Co$^{III}$ ↔ Co$^{III}$Co$^{III}$ ↔ Co$^{IV}$Co$^{III}$ ↔ Co$^{IV}$Co$^{IV}$ (see proposed detailed structural motifs in Supplementary Fig. 7). In contrast, Co$_3$O$_4$/CeO$_2$ displayed no obvious pre-OER redox features and a much lower onset potential for acidic OER (Fig. 2a and Supplementary Fig. 8b), suggesting the redox properties of Co$_3$O$_4$ can be effectively regulated by the introduction of CeO$_2$. Note that CeO$_2$ itself shows no redox feature and very poor activity toward OER in acid (Supplementary Fig. 9). The Co$_3$O$_4$/CeO$_2$ catalyst prepared by introducing a nominal 10 at% Ce metal content during the electrodeposition process exhibited the highest acidic OER catalytic performance (Supplementary Fig. 10) and was therefore studied in the rest of this work. The overpotentials required for Co$_3$O$_4$ and Co$_3$O$_4$/CeO$_2$ (10 at% Ce) to reach a geometric catalytic current density of 10 mA cm$^{-2}$ on FTO electrodes were 507 ± 5 and 423 ± 8 mV, respectively, showing a substantial improvement of ~84 mV after the introduction of CeO$_2$ (Fig. 2a inset). The Tafel slopes of the acidic OER on Co$_3$O$_4$ and Co$_3$O$_4$/CeO$_2$ were 110.8 and 88.1 mV dec$^{-1}$, respectively (Fig. 2c). Both are in the range of 60– 120 mV dec$^{-1}$, indicating a mixed kinetic control mechanism[44]. A second linear Tafel region was observed in Co$_3$O$_4$ (in the overpotential range of 350–425 mV shaded in pink), which originates from the charge-accumulation process due to the oxidation of dimeric Co$^{IV}$Co$^{III}$ to Co$^{IV}$Co$^{IV}$. In contrast, Co$_3$O$_4$/CeO$_2$ only exhibits a single linear Tafel region with a smaller slope of 88.1 mV dec$^{-1}$, which suggests that the OER catalytic onset takes place at a much lower overpotential of ~300 mV without noticeable charge-accumulation of dimeric Co redox centers.

The intrinsic acidic OER catalytic activities of Co$_3$O$_4$ and Co$_3$O$_4$/CeO$_2$ catalysts on FTO electrodes were further extracted based on double-layer capacitance ($C_{dl}$) measurements and electrochemically active surface area (ECSA) normalization. The $C_{dl}$ values of Co$_3$O$_4$ (7.31 mF cm$^{-2}$) and Co$_3$O$_4$/CeO$_2$ (23.26 mF cm$^{-2}$) (Supplementary Fig. 11) showed that the introduction of CeO$_2$ substantially increased the ECSA. Nevertheless, after normalizing the geometric catalytic current density by the ECSA derived from $C_{dl}$ (see Methods for details)[45], Co$_3$O$_4$/CeO$_2$ still displayed a much lower OER catalytic onset potential than Co$_3$O$_4$

and a much higher ECSA-normalized catalytic current density of 23.7 μA cm$^{-2}$ at the overpotential of 450 mV, which doubled that of Co$_3$O$_4$ at the same overpotential (Fig. 2d). These results confirm that Co$_3$O$_4$/CeO$_2$ features enhanced intrinsic OER catalytic activity compared to Co$_3$O$_4$ in acidic media.

We further examined the electron transfer kinetics of Co$_3$O$_4$ and Co$_3$O$_4$/CeO$_2$ catalysts on FTO electrodes using electrochemical impedance spectroscopy (EIS) at different potentials and extracted the charge transfer resistance ($R_{ct}$) of the catalytic OER from EIS fitting using the Voigt circuit model (Supplementary Fig. 12 and Supplementary Table 2)[46]. At the potentials between 1.566 and 1.616 V vs. reversible hydrogen electrode (RHE), the charge accumulation process due to the oxidation of dimeric Co redox centers dominated on the Co$_3$O$_4$ catalyst, whereas the catalytic OER already took place on the Co$_3$O$_4$/CeO$_2$ catalyst. As a result, the $R_{ct}$ values of Co$_3$O$_4$ were one order of magnitude higher than those of Co$_3$O$_4$/CeO$_2$ (Supplementary Table 2). Once OER dominated on Co$_3$O$_4$ after the oxidation of dimeric Co$^{IV}$Co$^{III}$ to Co$^{IV}$Co$^{IV}$ at the higher potential of 1.716 V vs. RHE, its $R_{ct}$ substantially decreased to be on the same order of magnitude as that of Co$_3$O$_4$/CeO$_2$ (Supplementary Table 2). These EIS results suggest that the catalytic OER on Co$_3$O$_4$ takes place efficiently only after overcoming the sluggish kinetic step associated with the charge accumulation process to form dimeric Co$^{IV}$Co$^{IV}$, and the introduction of CeO$_2$ effectively regulates the redox properties of Co$_3$O$_4$ and substantially enhances the electron transfer kinetics of the catalytic OER at a much lower overpotential.

We further verified that the enhanced catalytic activity of Co$_3$O$_4$/CeO$_2$ could not be attributed to the decreased crystallinity of Co$_3$O$_4$ due to the introduction of CeO$_2$ (see earlier discussions of Fig. 1e and Supplementary Fig. 5). By varying the annealing temperature, a series of Co$_3$O$_4$ and Co$_3$O$_4$/CeO$_2$ samples with different degrees of crystallinity were prepared (Supplementary Fig. 13). The pre-OER redox features were consistently present in Co$_3$O$_4$ and absent in Co$_3$O$_4$/CeO$_2$ regardless of different annealing temperatures, suggesting the redox properties of Co$_3$O$_4$ are unaffected by the degree of crystallinity (Supplementary Fig. 14a). Moreover, in contrast to Co$_3$O$_4$ that appeared to be

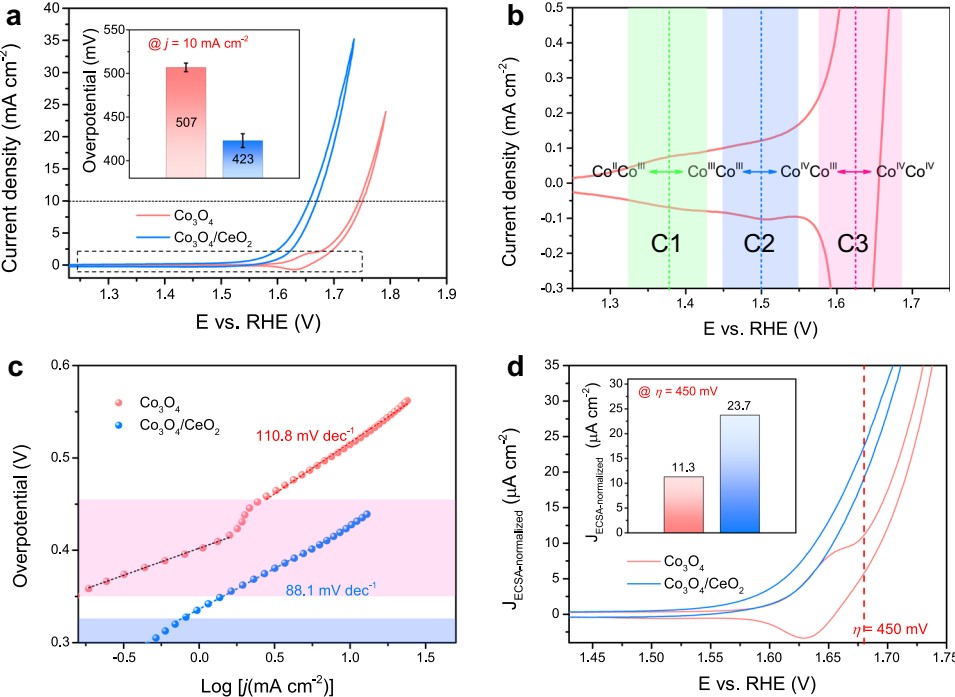

**Fig. 2 Electrochemical characterizations of Co$_3$O$_4$ and Co$_3$O$_4$/CeO$_2$ (prepared with 10 at% Ce) catalysts on FTO electrodes in 0.5 M H$_2$SO$_4$ solution. a** *iR*-corrected CV curves of both catalysts, the inset shows the overpotential (with error bar) required for each catalyst to reach a geometric catalytic current density of 10 mA cm$^{-2}$ based on the averages of three individual electrodes. **b** Magnified CV curve of the Co$_3$O$_4$ catalyst that highlights the three pre-OER redox features and the corresponding C1, C2, and C3 cathodic peaks. **c** The corresponding Tafel plots of both catalysts. **d** ECSA-normalized CV curves of both catalysts, the inset shows the ECSA-normalized catalytic current density ($J_{ECSA-normalized}$) of each catalyst at the overpotential of 450 mV.

more active when less crystalline, the OER activity of Co$_3$O$_4$/CeO$_2$ remained nearly constant regardless of the different sample crystallinity (Supplementary Fig. 14c, d), indicating the catalytic activity enhancement in Co$_3$O$_4$/CeO$_2$ originates from the regulated redox properties rather than sample crystallinity.

To shed light on the pre-OER redox mechanisms of Co$_3$O$_4$ and understand their relationships to the catalytic activity, we conducted pH-dependence analysis of the C3 peak on the Co$_3$O$_4$ catalyst in H$_2$SO$_4$ solution in the pH range of 0.48–1.24 (Fig. 3a and Supplementary Fig. 15a). The peak potential vs. standard hydrogen electrode was plotted against the solution pH (Fig. 3a inset). The slope of 95.9 ± 4.8 mV per pH unit suggests a 2 e$^-$/3 H$^+$ coupled redox process[47], which is different from the 59 or 120 mV per pH unit expected for a 1 e$^-$/1 H$^+$ or 1 e$^-$/2 H$^+$ process, respectively[48]. In addition, CV curves of Co$_3$O$_4$ recorded at different scan rates in 0.5 M H$_2$SO$_4$ solution (Fig. 3b and Supplementary Fig. 16) reveal the first-order power law relationship between the three cathodic peak current densities and the scan rate (Fig. 3b inset), suggesting that the C3 peak is associated with a surface capacitive process[49,50]. Thus, this crucial third redox feature of Co$_3$O$_4$ corresponds to a 2 e$^-$/3 H$^+$ surface capacitive process of Co$^{IV}$Co$^{III}$ ↔ $^{IV}$Co$^{IV}$, consistent with the proposed structural motifs in Supplementary Fig. 7. Moreover, this prominent 2 e$^-$/3 H$^+$ redox feature of Co$_3$O$_4$ also indicates that the dimeric Co$^{IV}$Co$^{IV}$ intermediate is partially stabilized and therefore cannot undergo a rapid catalytic turnover process to produce O$_2$ and return to the lower valence resting states[34,51], thus resulting in an increased overpotential to drive the catalytic reaction[35,36]. In contrast, the absence of this pre-OER redox feature in Co$_3$O$_4$/CeO$_2$ suggests that the introduction of CeO$_2$ effectively destabilizes the dimeric Co$^{IV}$Co$^{IV}$ intermediate and accelerates the catalytic turnover process, which leads to the enhanced acidic OER activity of the nanocomposite catalyst.

Since the oxygen source for acidic OER is H$_2$O, the cleavage of HO-H bond and the proton transfer properties are important factors that could affect the catalytic activity, similar to the case of alkaline HER[52]. Therefore, we collected the CV curves of both Co$_3$O$_4$ and Co$_3$O$_4$/CeO$_2$ catalysts on FTO electrodes in the protonic (0.5 M H$_2$SO$_4$ in H$_2$O) vs. deuteric (0.5 M D$_2$SO$_4$ in D$_2$O) solution to investigate the KIE of acidic OER (Fig. 3c and Supplementary Fig. 17). Substituting proton with deuterium affects both the thermodynamics and the kinetics of reactions involving protons[34]. The shift of 33 mV in the standard equilibrium potential of the OER when proton is exchanged with deuterium [1.229 V vs. RHE for O$_2$/H$_2$O as opposed to 1.262 V vs. reversible deuterium electrode (RDE) for O$_2$/D$_2$O] is attributed to the change in the reaction thermodynamics (Fig. 3c)[34,53]. To separate the KIE from the reaction thermodynamics, linear sweep voltammetry curves were presented on the overpotential scale, and the KIE value was calculated based on the catalytic current density in the protonic vs. deuteric solution at the same overpotential (Fig. 3d, also see Methods for details). For both catalysts, the KIE values in OER potential regions fluctuated around the upper limit of secondary KIE (~1.5) with the absence of primary KIE, indicating that proton transfer is not rate-limiting for the acidic OER on both catalysts[34,53]. In addition, the pH-dependence analysis of the catalytic current densities at fixed overpotentials showed that the reaction order with respect to pH is close to zero on the RHE scale for acidic OER on both catalysts (Supplementary Fig. 15), indicating the catalytic reaction is less dependent on the proton concentration in the electrolyte for both catalysts. These results suggest that the enhanced acidic OER activity of Co$_3$O$_4$/CeO$_2$ is unrelated to the proton transfer properties of the nanocomposite.

We further conducted temperature-dependent kinetic analysis of both Co$_3$O$_4$ and Co$_3$O$_4$/CeO$_2$ catalysts to extract the apparent activation energy ($E_{app}$) and pre-exponential factor ($A_{app}$) of the

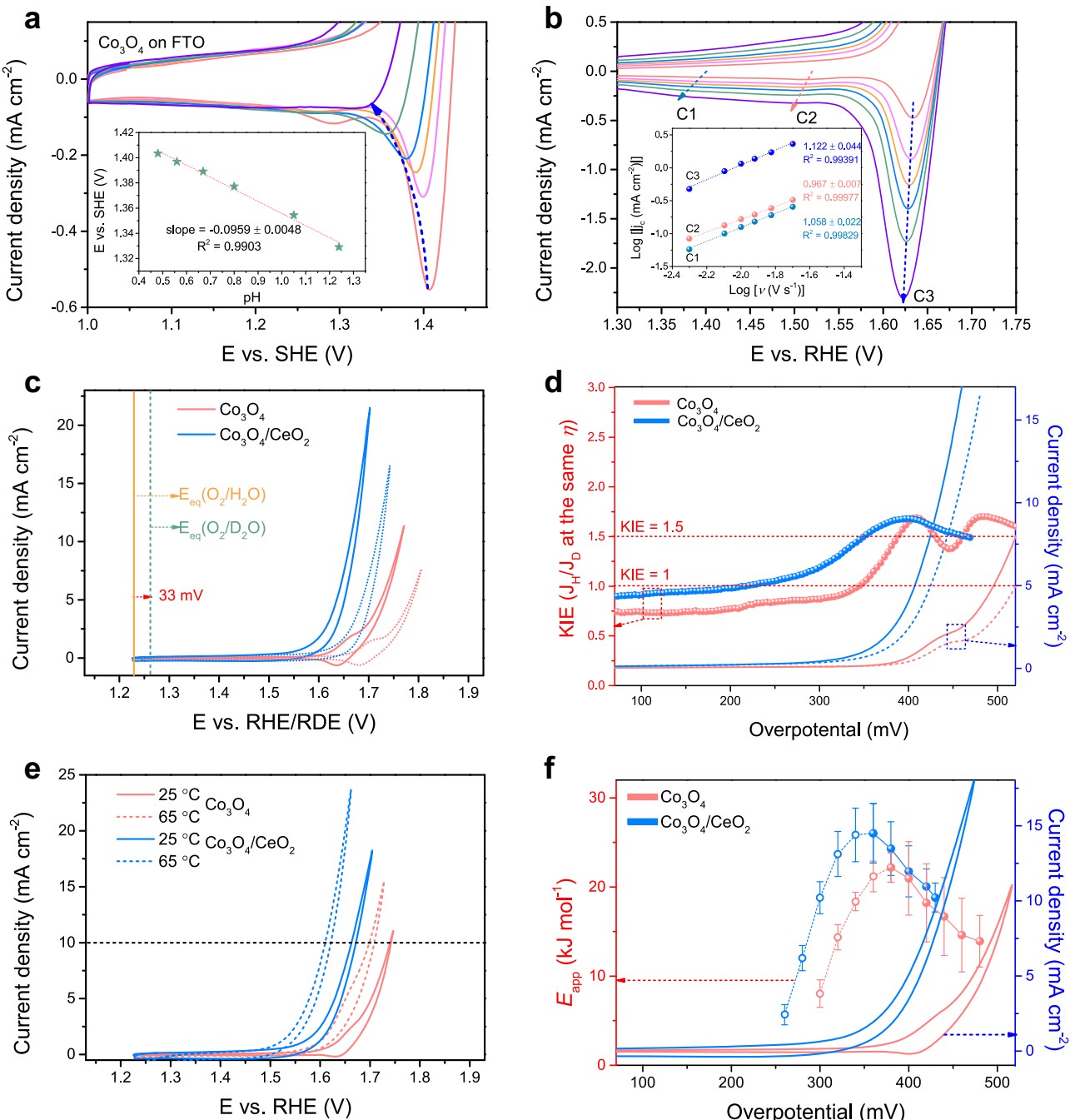

**Fig. 3 The pH-dependence, kinetic isotope effect (KIE) and apparent activation energy ($E_{app}$) analyses of the acidic OER on $Co_3O_4$ and $Co_3O_4/CeO_2$ catalysts on FTO electrodes. a** CV curves of $Co_3O_4$ recorded in $H_2SO_4$ solutions with different pH values, the inset shows the C3 peak potential vs. SHE plotted against the solution pH. **b** CV curves of $Co_3O_4$ recorded at different scan rates in 0.5 M $H_2SO_4$ solution, the inset shows the logarithm of cathodic peak current density ($j_c$) plotted against the logarithm of scan rate ($\nu$). **c** CV curves of both catalysts recorded in 0.5 M $H_2SO_4$ in $H_2O$ solution on the RHE scale (solid) vs. in 0.5 M $D_2SO_4$ in $D_2O$ solution on the RDE scale (dashed). **d** The KIE curves plotted with the LSV curves adapted from (**c**) but presented on the overpotential scale. **e** CV curves of both catalysts recorded in 0.5 M $H_2SO_4$ solution at 25 vs. 65 °C. **f** The corresponding $E_{app}$ data point and error bar are calculated from CV curves recorded at different temperatures (see Supplementary Fig. 18 for details).

acidic OER and to examine how the introduction of $CeO_2$ affects the catalytic mechanism. CV curves of both catalysts on FTO electrodes were recorded in 0.5 M $H_2SO_4$ solution in the temperature range of 25–65 °C (Supplementary Fig. 18). As expected, the catalytic performances of both catalysts increased at elevated temperatures (Fig. 3e and Supplementary Fig. 18). The $E_{app}$ values of both catalysts at fixed overpotentials were calculated from the Arrhenius equation (Fig. 3f and

Supplementary Fig. 19, also see Methods for details)[54,55]. To completely capture the potential-dependent evolution of $E_{app}$, the analysis was performed both below and above the catalytic onset potential. On both catalysts, the $E_{app}$ value reached its maximum around the respective catalytic OER onset potential (Fig. 3f), consistent with the fact that $Co_3O_4/CeO_2$ requires a lower overpotential than $Co_3O_4$ to catalyze the OER. Interestingly, the $E_{app}$ values on both catalysts were very similar after the catalytic

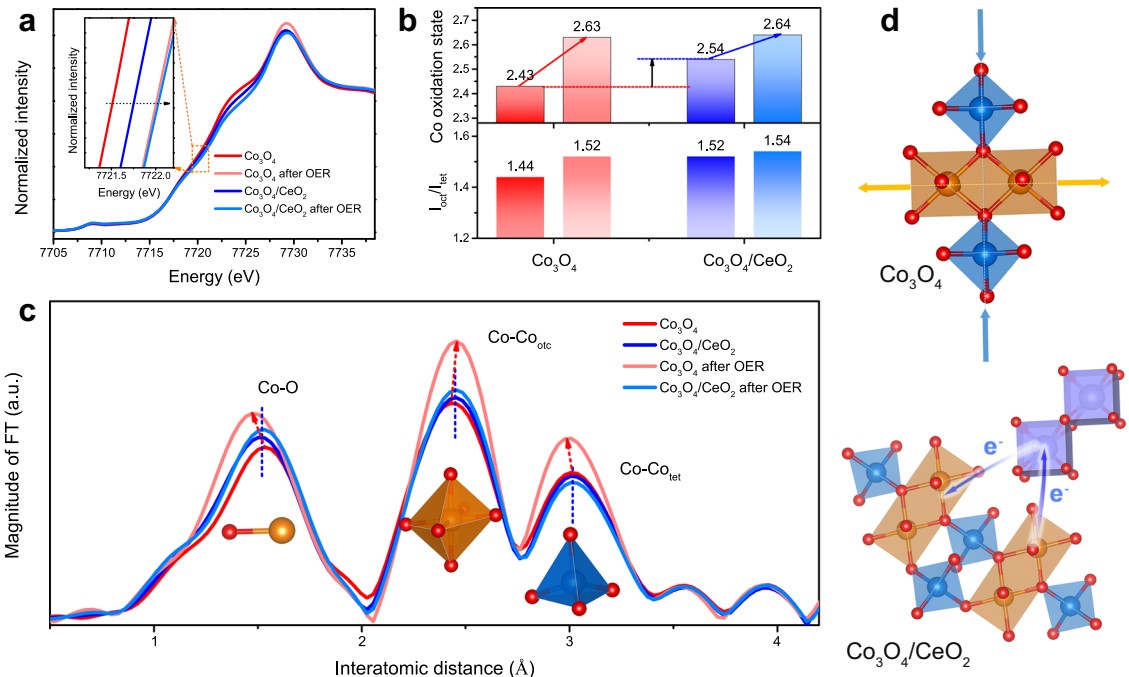

**Fig. 4 XAS characterizations of Co₃O₄ and Co₃O₄/CeO₂ catalysts before and after OER testing in 0.5 M H₂SO₄ solution to reveal the structural and oxidation state differences between the two catalysts. a** Co K-edge XANES spectra, the inset shows the upshift in the absorption edge energy after OER testing. **b** The average Co oxidation states and the intensity ratios of Co-Co$_{oct}$ and Co-Co$_{tet}$ scattering paths (I$_{oct}$/I$_{tet}$) of both catalysts. For each catalyst, the left and right columns represent the values before and after OER testing, respectively. **c** Fourier transforms (FT) of k³-weighted Co K-edge EXAFS spectra for both catalysts before and after OER testing. **d** Schematic illustrations of the local bonding environment changes in Co₃O₄ before and after OER testing and the hypothesized electronic modifications in Co₃O₄/CeO₂.

onsets (Fig. 3f), while more obvious differences are observed in the $A_{app}$ (Supplementary Fig. 20). The similar $E_{app}$ suggests that the introduction of CeO₂ does not alter the rate-determining step and the kinetic barrier for the formation of reaction intermediates, but rather enhances the intrinsic activity of the same type of catalytic active site in Co₃O₄ by modifying the entropy of activation (i.e., the number of active intermediates that enter the rate-determining step) and the interfacial concentration of active sites, as higher $A_{app}$ is extracted for Co₃O₄/CeO₂ at the same overpotential[56–58]. Therefore, these KIE, pH- and temperature-dependence analyses exclude several other factors, so we attribute the enhanced acidic OER activity to the regulation of the redox properties in Co₃O₄/CeO₂ resulted from the modified local bonding environment, as explained below.

**Spectroscopic characterization of the structural evolution in Co₃O₄/CeO₂.** We performed ex situ XAS on Co₃O₄ and Co₃O₄/CeO₂ catalysts before and after OER testing in 0.5 M H₂SO₄ solution to understand the their structural evolution. Scanning electron microscopy (SEM)-EDS and XPS analyses confirmed that the elemental compositions of Co₃O₄/CeO₂ were mostly preserved after OER testing (Supplementary Figs. 21 and 22 and Supplementary Table 1). The surface-sensitive XPS revealed no obvious shift in the binding energies of the Co 2p signals after the introduction of CeO₂ (Supplementary Fig. 22a, d). Ultraviolet photoelectron spectroscopy (UPS) (Supplementary Fig. 23) showed larger work function in Co₃O₄/CeO₂ than pure Co₃O₄, suggesting the electronic structure in Co₃O₄/CeO₂ was slightly modified due to possible electronic interactions between Co₃O₄ and CeO₂. XAS is more sensitive to subtle changes in the oxidation states and the local bonding environments throughout the nanocomposite samples. According to the relative absorption edge positions in the Co K-edge X-ray absorption near-edge

spectra (Fig. 4a), the Co₃O₄/CeO₂ exhibited a slightly higher Co oxidation state than the as-synthesized Co₃O₄, and the Co oxidation states in both catalysts increased and became similar after OER testing (inset of Fig. 4a). The absorption edge energies were further determined by an integral method[59] and the average Co valence states were calculated (see Methods for details)[34,60]. The average Co oxidation states in the as-synthesized Co₃O₄ and Co₃O₄/CeO₂ were 2.43 and 2.54, respectively; but after OER testing, both were raised to comparable higher values of 2.63 and 2.64 (upper panel of Fig. 4b). Therefore, although the introduction of CeO₂ slightly increased the Co oxidation state in the Co₃O₄/CeO₂ catalyst, such difference did not persist after OER testing and therefore might not directly account for the distinct electrochemical properties of Co₃O₄/CeO₂ vs. Co₃O₄. Moreover, a comparison of various Co₃O₄ samples annealed at different temperatures also suggests that a higher Co oxidation state before OER testing (Supplementary Fig. 24) does not necessarily result in changes in the pre-OER redox features (Supplementary Fig. 14a).

Besides the higher Co oxidation state, the changes in local bonding environment of Co₃O₄ induced by CeO₂ were also observed, as revealed by extended X-ray absorption fine structure (EXAFS) (Fig. 4c and Supplementary Fig. 25). Fourier transforms of k³-weighted Co K-edge EXAFS spectra of both Co₃O₄ and Co₃O₄/CeO₂ catalysts displayed three major signals associated with the Co-O, Co-Co$_{oct}$ (octahedral site), and Co-Co$_{tet}$ (tetrahedral site) scattering paths (Fig. 4c). Compared to the as-synthesized Co₃O₄ (Fig. 4c red trace), a shorter Co-O bond distance was observed in the Co₃O₄/CeO₂ (Fig. 4c blue trace) due to the higher positive charge density at the Co centers[61] after the electron redistribution from Co₃O₄ to CeO₂, as illustrated in the bottom scheme in Fig. 4d. More importantly, the bond distances in Co₃O₄/CeO₂ remained the same after OER testing (Fig. 4c light blue trace), and the crystal structure barely changed,

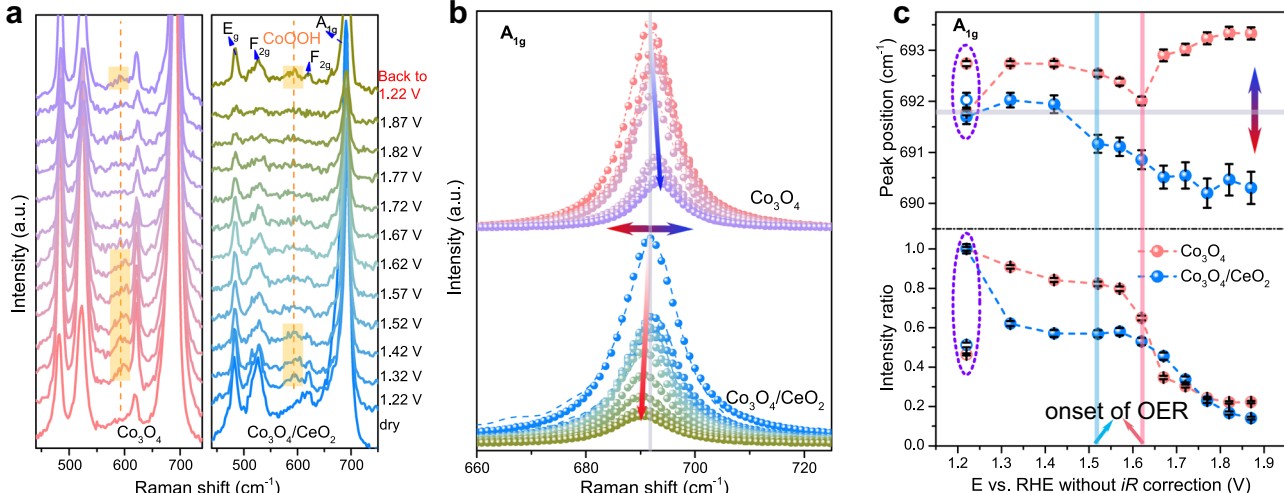

**Fig. 5 In situ Raman characterizations of Co$_3$O$_4$ and Co$_3$O$_4$/CeO$_2$ catalysts on carbon paper electrodes during OER testing in 0.5 M H$_2$SO$_4$ solution to reveal the structural evolution of catalysts. a** The in situ Raman spectra of Co$_3$O$_4$ (left panel) and Co$_3$O$_4$/CeO$_2$ (right panel) at various constant potentials (vs. RHE) without *iR* correction (increased from 1.22 to 1.87 V and then back to 1.22 V). The Raman spectra of the dry samples were also presented at the bottom for comparisons. **b** The Raman A$_{1g}$ peaks of Co$_3$O$_4$ (top) and Co$_3$O$_4$/CeO$_2$ (bottom) were fitted with Lorentzian function to extract the peak positions, intensity, and FWHM (dash lines: raw spectra; dots: fitting results). **c** The Raman A$_{1g}$ peak positions (upper panel) and intensity ratio with respect to the initial intensity at 1.22 V (lower panel) plotted against the applied potential. The open symbols represent the data collected at 1.22 V at the end after applying the higher potential sequence. The error bar represents the error from fitting.

as shown by the identical intensity ratio of Co-Co$_{oct}$ and Co-Co$_{tet}$ scattering paths (I$_{oct}$/I$_{tet}$) before and after OER testing (lower panel of Fig. 4b). In contrast, there were distinct changes in the bonding distances in Co$_3$O$_4$ after OER reaction (Fig. 4c light red curve), namely the shortening of both Co-O and Co-Co$_{tet}$ bonds and the elongation of Co-Co$_{oct}$ bond, as illustrated in the top scheme in Fig. 4d. Moreover, the I$_{oct}$/I$_{tet}$ ratio in Co$_3$O$_4$ displayed an obvious increase from 1.44 to 1.52 after OER testing (lower panel of Fig. 4b), suggesting the crystal structure of Co$_3$O$_4$ underwent dynamic changes during OER reaction, as revealed by the prominent three sets of pre-OER redox features, which might be similar to the formation of active structure motifs during OER reactions in alkaline or neutral media[26,29].

**In situ Raman studies of the OER reaction mechanisms**. To further unveil the relationships between the catalytic activity enhancement, redox-mediated surface reconstruction, and the modified local bonding environment in Co$_3$O$_4$/CeO$_2$ nanocomposites, we conducted in situ Raman experiments on both catalysts in 0.5 M H$_2$SO$_4$ solution under OER conditions (Supplementary Fig. 26). Both dry samples of Co$_3$O$_4$ and Co$_3$O$_4$/CeO$_2$ display four characteristic Raman peaks corresponding to the E$_g$ (~480 cm$^{-1}$), F$_{2g}$ (~520 cm$^{-1}$), F$_{2g}$ (~620 cm$^{-1}$), and A$_{1g}$ (~690 cm$^{-1}$) phonon modes of Co$_3$O$_4$ spinel oxides (Fig. 5a)[62]. After the samples were immersed in the electrolyte, another Raman signal emerged at ~600 cm$^{-1}$ at the applied potential of 1.22 V (vs. RHE), which was attributed to the formation of CoOOH species at the surface[31]. This CoOOH species was less clearly detected at high potentials and started to disappear from the Co$_3$O$_4$/CeO$_2$ and Co$_3$O$_4$ surfaces at 1.52 and 1.62 V vs. RHE, respectively, which coincided with their respective OER onset potentials (Supplementary Fig. 26), as well as the two pre-OER redox features of Co$_3$O$_4$ associated with Co$^{III}$Co$^{III}$ ↔ Co$^{III}$Co$^{IV}$ (~1.50 V vs. RHE) and Co$^{III}$Co$^{IV}$ ↔ Co$^{IV}$Co$^{IV}$ (~1.63 V vs. RHE) transitions (Fig. 2b). Clearly, this CoOOH species is not the actual active phase for acidic OER and needs to be further oxidized into Co$^{IV}$ species. The disappearance of this CoOOH species from Co$_3$O$_4$/CeO$_2$ at a lower potential indicates that it is easier to oxidize the active Co sites in the Co$_3$O$_4$/CeO$_2$ catalyst into OER-

active Co$^{IV}$ species compared to those Co sites in the pure Co$_3$O$_4$. The intensities of all Raman peaks at higher applied potentials decrease substantially (Fig. 5b, c lower panel), which was usually accompanied with the increase in average valence state of Co atoms[63]. When the applied potential was finally switched back from 1.87 to 1.22 V vs. RHE, the peak intensities partially recovered (lower panel in Fig. 5c) and the CoOOH species was clearly detected again.

To understand the evolution of the local bonding environments at the catalyst surface during the OER process, the peak position, intensity, and full width at half maximum (FWHM) of the Raman A$_{1g}$ peak (~690 cm$^{-1}$) were extracted by fitting with Lorentzian function (Fig. 5b, c). The shift in the peak position as a function of applied potential can be interpreted as either the change in crystallinity (e.g., red-shift with broadening in FWHM happens when the crystallinity decreases dramatically), or the generation of strain/stress (i.e., lattice contraction/extension)[64,65]. Since the marginal variations in the peak FWHM suggested the crystalline domain sizes of both samples remain relatively constant during the OER process (Supplementary Fig. 27), the observed peak position shift should result from the lattice contraction/extension and surface reconstruction due to the changing local bonding environments. More importantly, the peak positions shift in opposite directions on these two catalysts as the potential goes over the OER catalytic onsets (Fig. 5c upper panel). Co$_3$O$_4$/CeO$_2$ showed a red-shift in the A$_{1g}$ peak position after the onset of OER at 1.52 V vs. RHE. Red-shifts in Raman signals are commonly observed in OER catalysts (CoO$_x$[63,66], NiOOH[67], NiFe, and CoFe oxyhydroxides[68]) at OER operating potentials, and they generally reflect the characteristic vibration for local bonding environment at the outer layer of catalysts with oxidized active site during OER. Thus, the generation of active Co$^{IV}$ species that can participate in a fast and efficient OER process should lead to the observed red-shift of the Raman signals. In contrast, blue-shifts in Raman signals usually suggest lattice contraction and charge redistribution[64,69]. Unlike the more active Co$_3$O$_4$/CeO$_2$, the pure Co$_3$O$_4$ catalyst would go through substantial charge-accumulation surface reconstruction (Co$^{III}$Co$^{IV}$ ↔ Co$^{IV}$Co$^{IV}$) at ~1.62 V around the onset for OER. The Co$^{IV}$ species generated during this process are stabilized and cannot

participate in fast OER turnover since the reduction peak could be still observed when the potential was scanned backwards, thus they lead to a blue-shift in the Raman signals (Fig. 5c). Another interesting difference is that the peak position of $Co_3O_4/CeO_2$ at 1.22 V vs. RHE remains almost unchanged before and after applying the higher potential sequence, suggesting the flexibility in the local bonding environment of $Co_3O_4$ in the composite catalyst. However, the peak position of $Co_3O_4$ cannot fully recover after the same potential cycle, with the final peak at ~1 cm$^{-1}$ higher in wavenumber accordingly (Fig. 5c upper panel and Supplementary Fig. 28), which is consistent with the positive charge accumulated at the Co center with shorter Co-O bond in the $Co_3O_4$ sample after OER (Fig. 4a–c). Together with the ex situ XAS results, the in situ Raman results clearly demonstrate that the bonding environment surrounding Co centers is modified in the $Co_3O_4/CeO_2$ catalyst, which allows the active Co sites to be more readily oxidized and avoid the substantial potential-determining surface reconstruction that would otherwise form stabilized dimeric $Co^{IV}Co^{IV}$ with charge accumulation and lattice contraction. As $Co^{IV}$ is the key intermediate to start OER process, the more facile formation of $Co^{IV}$ species and destabilization of $Co^{IV}Co^{IV}$ in $Co_3O_4/CeO_2$ would allow faster OER kinetics thus enhance the catalytic activity.

**Electrode performance and stability of $Co_3O_4/CeO_2$ nanocomposites**. We further optimized the overall electrode performance by replacing the FTO substrate with high-surface-area three-dimensional carbon paper substrate that facilitates electron and ion transport and gas bubble release. To reach a geometric catalytic current density of 10 mA cm$^{-2}$ in 0.5 M $H_2SO_4$ solution, $Co_3O_4/CeO_2$ on carbon paper electrode only required an overpotential as low as 347 mV, which is only 46 mV higher than that needed for the benchmark $RuO_2$ catalyst on carbon paper electrode (Supplementary Fig. 29). A comprehensive comparison shows that $Co_3O_4/CeO_2$ is an efficient earth-abundant metal oxide-based electrocatalysts reported to date for the acidic OER (Supplementary Table 3).

Lastly, we examined the acidic OER stability of the $Co_3O_4/CeO_2$ catalyst, since the tradeoff between activity and stability has usually been observed in acidic OER catalysts[15,16]. As discussed earlier, the apparent elemental compositions of $Co_3O_4/CeO_2$ changed little after the OER test (Supplementary Figs. 21 and 22). Since it is known that $Co_3O_4$ dissolves very slowly under acidic OER conditions based on detection of metal leaching[23], we used ICP-MS to monitor the catalyst dissolution rate of the high-performance $Co_3O_4/CeO_2$ on carbon paper electrode during long-term chronopotentiometry tests at 10 mA cm$^{-2}$ in 0.5 M $H_2SO_4$ solutions (Supplementary Fig. 30). $Co_3O_4/CeO_2$ displayed essentially the same rate of potential increase over time as $Co_3O_4$ in 0.5 or 0.05 M $H_2SO_4$ solution over 50 or 100 h continuous operation, respectively (Supplementary Fig. 30a, c). The cobalt dissolution rate of $Co_3O_4/CeO_2$ also coincided with that of $Co_3O_4$ in 0.5 M $H_2SO_4$ solution (Supplementary Fig. 30b). The metal dissolution rates of both catalysts were also investigated under open circuit condition without an applied bias (Supplementary Fig. 31). Both catalysts showed inferior stability under open circuit condition compared to their respective stability under anodically biased OER condition, suggesting that the applied bias is important for the long-term stability of earth-abundant Co oxides during acidic OER operation[70]. It is noteworthy that $Co_3O_4/CeO_2$ displayed no obvious Ce dissolution and much slower Co dissolution than pure $Co_3O_4$ under open circuit condition. Thus, the more active $Co_3O_4/CeO_2$ exhibits a comparable OER stability but an enhanced open circuit stability compared to the less active $Co_3O_4$, and therefore breaks the activity/stability tradeoff.

## Discussion

In conclusion, $Co_3O_4/CeO_2$ nanocomposite is established as an active earth-abundant OER electrocatalyst in acidic media. The overpotentials required for $Co_3O_4/CeO_2$ to achieve a geometric catalytic current density of 10 mA cm$^{-2}$ on FTO and carbon paper electrodes are ~423 and 347 mV, respectively, making it an efficient earth-abundant electrocatalysts for acidic OER. In-depth electrochemical characterizations using the KIE, pH-, and temperature-dependence analyses, together with in situ Raman and ex situ XAS structural characterizations of the $Co_3O_4/CeO_2$ catalyst before and after OER testing, consistently reveal the microstructural states of the catalysts and their changes through the OER processes. The introduction of nanocrystalline $CeO_2$ modifies the electronic structures and creates a more favorable local bonding environment in $Co_3O_4$ that allows the $Co^{III}$ surface species to be easily oxidized into OER-active $Co^{IV}$ species and suppresses the charge accumulation of $Co_3O_4$ under electrochemical conditions, which are the keys to bypassing the potential-determining redox step in $Co_3O_4$ that result in substantial surface reconstruction and thus enhancing the acidic OER activity. Interestingly, $Co_3O_4/CeO_2$ also breaks the activity/ stability tradeoff by featuring enhanced activity but comparable acidic OER stability and better open circuit stability in comparison with $Co_3O_4$. We hope these findings could stimulate future studies to further elucidate the active site structures and the catalytic mechanisms of nanocomposite OER catalysts using other in situ and/or operando techniques. This work not only establishes an active earth-abundant nanocomposite catalyst ($Co_3O_4/CeO_2$) for OER in acidic media, but also stimulates mechanistic understandings and provides an effective strategy to design more efficient and stable nanocomposite electrocatalysts for acidic OER or other reactions in the future.

## Methods

**Chemicals**. All chemicals were purchased from Sigma-Aldrich and used as received without further purification, unless noted otherwise. Deionized nanopure water (18.2 MΩ·cm) from a Thermo Scientific Barnstead water purification system was used for all experiments.

**Synthesis of $Co_3O_4$ and $Co_3O_4/CeO_2$ on FTO or carbon paper**. The corresponding metal hydroxide precursors were first synthesized on the substrates by electrodeposition from a solution of the corresponding metal nitrate(s) with a total concentration of 0.1 molar (mol). For synthesizing the Ce-doped $Co(OH)_2$ precursor, 10 mol percent (mol%) of $Co(NO_3)_2$ in the solution was replaced with Ce $(NO_3)_3$. Note that the as-received carbon paper substrate (Fuel Cell Earth, TGP-H-060) was Teflon-coated; therefore, it was first treated with oxygen plasma at 300 W power for 15 min for each side and then annealed in air at 700 °C for 1 h to make the surface hydrophilic. Prior to the electrodeposition, the FTO and carbon paper substrates were successively washed with acetone, ethanol, and nanopure water. During the electrodeposition, an Ag/AgCl reference electrode and a Pt mesh counter electrode were used, and a constant potential of –1.0 V vs. Ag/AgCl was applied on the substrates for 3 and 10 min in the case of FTO and carbon paper, respectively. During the electrodeposition, the reduction of nitrate generated OH⁻ and a local alkaline environment near the substrate, and subsequently metal hydroxides were formed on the substrate[71]:

$$NO_3^- + 7\,H_2O + 8\,e^- \rightarrow NH_4^+ + 10\,OH^- \tag{1}$$

$$Co^{2+} + 2\,OH^- \rightarrow Co(OH)_2 \tag{2}$$

After the electrodeposition, the metal hydroxide precursors were dried at 80 °C for 6 h, and then annealed in air at 400 °C (or 300 or 500 °C as specifically discussed) for 2 h in a muffle furnace to transform into oxides.

**Structural characterizations**. SEM and EDS were conducted on a Zeiss Supra 55VP field emission SEM equipped with a Thermo Fisher Scientific UltraDry EDS detector. The accelerating voltage for SEM and EDS were 3 and 15 kV, respectively. Transmission electron spectroscopy images and elemental mappings were collected using a JEM-2100F microscope equipped with an Oxford energy-dispersive X-ray analysis system, with the accelerating voltage of 200 kV. PXRD was performed on a Bruker D8 Advance powder X-ray diffractometer using Cu Kα radiation. XPS was performed on a Thermo Scientific K-Alpha XPS system with an Al Kα X-ray source. UPS was collected on a Thermo ESCALAB 250Xi XPS system with a He I

source gun. The Raman spectra were collected on a Thermo Fisher Scientific DXRxi Raman imaging microscope with a 532 nm laser. The ICP-MS analysis was carried out on a Shimadzu ICPMS-2030 spectrometer. The XAS were collected in the transmission mode at the Advanced Photon Source Beamline 10-BM-B at the Argonne National laboratory. To collect the Co K-edge in the energy window from 7.450 to 8.650 keV, a 71/29 $N_2$/He gas mixture was used in the $I_0$ ion chamber to achieve 10% absorption, while a 68/32 $N_2$/Ar gas mixture was used in the $I_t$ ion chamber to achieve 70% absorption (calculated using Hephaestus at an energy of 7.709 keV). The Co foil standard was used for the energy calibration.

**Electrochemical measurements.** All electrochemical measurements were conducted in a conventional three-electrode setup using a Bio-Logic SP-200 potentiostat. The $Co_3O_4$ or $Co_3O_4$/$CeO_2$ catalyst grown on FTO or carbon paper was directly used as the working electrode, along with an Ag/AgCl reference electrode and a Pt mesh counter electrode in 0.5 M $H_2SO_4$ solution. CV was performed at the scan rate of 5 mV s$^{-1}$. EIS was collected in the frequency range from 100 kHz to 50 mHz. All CV curves were manually iR-corrected based on EIS results. To extract the double-layer capacitance ($C_{dl}$), CV was collected in pre-OER potential region at various scan rates from 10 to 60 mV s$^{-1}$. The relationship between ECSA (cm$^2$) and $C_{dl}$ (mF) is shown in Eq. (3):

$$\text{ECSA} = C_{dl}/C_s \quad (3)$$

where $C_s$ is general specific capacitance, which is a constant of 0.035 mF cm$^{-2}$ in the literature[45].

All potentials were reported versus the RHE using Eq. (4):

$$E(\text{RHE}) = E(\text{Ag/AgCl}) + 0.059\,\text{pH} + 0.197 \quad (4)$$

The operational stability of the catalyst was tested by running chronopotentiometry tests at a constant geometric catalytic current density of 10 mA cm$^{-2}$ in 0.5 (or 0.05) M $H_2SO_4$ solution for 50 (or 100) h.

**Reaction order with respect to pH.** To extract the reaction order with respect to pH for the acidic OER, the electrochemical measurements of the catalysts were conducted in $H_2SO_4$ solutions with different pH values. The reaction order with respect to pH was calculated using Eq. (5)[27,72]:

$$\text{Reaction order} = \left| \frac{\partial(\log_{10} j)}{\partial \text{pH}} \right|_\eta \quad (5)$$

where $j$ is the catalytic current density at a fixed overpotential $\eta$.

**Kinetic isotope effect (KIE).** To evaluate the KIE, the electrochemical measurements of the catalysts were conducted in both protonic (0.5 M $H_2SO_4$ in $H_2O$) and deuteric (0.5 M $D_2SO_4$ in $D_2O$) solutions. The pD value of the deuteric solution was determined by 0.41 plus the measured pH value using a glass membrane pH electrode connected to a pH meter[73]. The potential on RDE scale was calculated using Eq. (6):

$$E(\text{RDE}) = E(\text{Ag/AgCl}) + 0.059\,\text{pD} + 0.197 + 0.013 \quad (6)$$

where the term of +0.013 originates from the difference in the standard equilibrium potentials of the deuterium couple ($D_2$/$D^+$) and the proton couple ($H_2$/$H^+$)[53].

The overpotentials of the OER in the protonic and deuteric solution were determined by Eqs. (7) and (8), respectively[53]:

$$\eta = E(\text{RHE}) - 1.229\,\text{V} \quad (7)$$

$$\eta = E(\text{RDE}) - 1.262\,\text{V} \quad (8)$$

The KIE was calculated using Eq. (9):

$$\text{KIE} = \left| \frac{j_{H_2O}}{j_{D_2O}} \right|_\eta \quad (9)$$

where $j_{H_2O}$ and $j_{D_2O}$ are the catalytic current density in the protonic and deuteric solution, respectively, at the same overpotential ($\eta$)[72].

**Apparent activation energy.** To extract the apparent activation energy ($E_{app}$) for the acidic OER, the electrochemical measurements of the catalysts were conducted in 0.5 M $H_2SO_4$ solution at different temperatures. For heterogeneous electrocatalytic reaction, the current density can be expressed from apparent activation energy ($E_{app}$) in the Arrhenius Eq. (10)[56,57]:

$$j = A_{app} \exp\left(-\frac{E_{app}}{RT}\right) \quad (10)$$

where $A_{app}$ is the apparent pre-exponential factor, $R$ is the ideal gas constant (8.314 J K$^{-1}$ mol$^{-1}$), $T$ is the temperature in Kelvin (K). Therefore, $E_{app}$ can be

further calculated from fitting the slope of the Arrhenius plot using Eq. (11)[54,56]:

$$\left| \frac{\partial(\log_{10} j)}{\partial(1/T)} \right|_\eta = -\frac{E_{app}}{2.303\,\text{R}} \quad (11)$$

while the intercept of $\log_{10} j$ vs. $1/T$ plot is the logarithm of $A_{app}$[57].

**Average Co valence state.** The absorption edge energies of the XAS spectra were first determined by an integral method shown in Eq. (12)[59]:

$$E_{edge} = \frac{1}{\mu_2 - \mu_1} \int_{\mu_1}^{\mu_2} E(\mu)\,d\mu \quad (12)$$

where $\mu_1 = 0.15$ and $\mu_2 = 1$ are the lower and upper limit, respectively, of the normalized absorption intensity that are used for the integral. The average Co valence states were then calculated by fitting the absorption edge energies determined earlier into an experimental equation developed by Dau et al.[34,60]:

$$\text{Oxidation state} = \frac{1}{2.29}(E_{edge} - 7714.1\,\text{eV}) \quad (13)$$

## Data availability

The data that support the findings in the paper can be found in the Source Data. Additional data presented in the Supplementary Information are available from the corresponding author upon reasonable request. Source Data are provided with this paper.

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

## Acknowledgements

This work is partially supported by University of Wisconsin–Madison UW2020 Initiative and King Abdullah University of Science and Technology (KAUST) OSR-2017-CRG6-3453.02. J. Z. H. thanks the China Scholarship Council (CSC) for fellowship support. B. S. thanks Natural Science Foundation of China (NSFC) Grant No. 51672057, 52072085, and 51722205 for support. H. S., R. D. R., and S. J. also thank the support from US NSF CHE-1955074. This research used resources of the Advanced Photon Source (APS), a US Department of Energy (DOE) Office of Science User Facility operated for the DOE Office of Science by Argonne National Laboratory under Contract No. DE-AC02-06CH11357. The XAS experiments were performed at the APS Beamline 10-BM-B. The authors acknowledge use of facilities and instrumentation at the UW-Madison Wisconsin Centers for Nanoscale Technology partially supported by the NSF through the University of Wisconsin Materials Research Science and Engineering Center (DMR-1720415).

## Author contributions

J. Z. H., B. S., and S. J. designed the experiments. J. Z. H. carried out the synthesis of catalysts, morphological and structural characterizations, and electrochemical measurements. H. S. collected the XPS spectra. J. Z. H. and H. S. collected the in situ Raman data. J. Z. H., H. S., and R. D. R. collected the ex situ XAS data at Advanced Photon Source in Argonne National Laboratory. J. Z. H. and S. J. wrote the manuscript. H. S., R. D. R., J. C. H., X. W., and B. S. performed the analysis and revised the manuscript.

## Competing interests

The authors declare no competing interests.
