## [Peer Review File · Nature Communications]

REVIEWER COMMENTS

Reviewer #1 (Remarks to the Author):

This is an interesting manuscript on the enhancement on the activity of Co oxide as a water oxidation electrocatalyst in acidic media, when decorated with ceria. The authors performed multiple and complementary analyses, including consistent characterization and mechanistic studies, resulting in a very interesting manuscript that I think really deserves to be published in Nat Commun.

I have two important comments, though, that I hope the authors can address in a revised version:

1) The authors compare the activity of Co₃O₄/CeO₂ vs Co₃O₄ in their conditions. However, they do not compare their results in detail with previous reports on Co oxides as OER electrocatalyst in acidic media. I think it would help if the authors could comment on previous, important references such as J. Am. Chem. Soc, 136(8), 3304–3311 and some others to put their results into the proper context.

2) The authors are quite honest stating that this OER enhancement does not improve Co oxide stability in acid. However, I miss some explicit experiments to show the stability of the catalysts in acid at open circuit potential. If I understood it right, the authors explore the stability during OER, at 10 mA cm⁻². In these conditions, Co₃O₄ is much more stable than without a bias. This open circuit instability is also crucial.

Reviewer #2 (Remarks to the Author):

The manuscript by Huang et al. reports the effect of the addition of ceria nanocrystals into a nanostructured cobalt oxide electrocatalyst for the electrochemical water oxidation in 0.5 M sulfuric acid aqueous solutions. The material science part is relatively limited, the material is electrodeposited onto fluorine doped tin dioxide and characterized after calcination by powder x-ray diffraction and electron microscopy, giving only a superficial picture of the chemical and structural properties of the film. Especially, the high resolution electron micrographs are of relatively low quality (Figure 1c,d) and their analysis not sufficient (interplanar distances are measured in only one direction for each crystallite and large zones are poorly colored masking the real information of the image). The same applies for the EDX maps, which present a too low resolution to distinguish the individual Ce and Co-containing crystallites. From the data presented the most performing material is a mixture of sub-10 nm nanocrystals of Co₃O₄ and CeO₂ with a 10 weight% content.

On the other hand, the electrochemical part is very well detailed and probably carried out following state of the art procedures (not being an expert, I can not properly judge the electrochemical and largest part of the manuscript).

The manuscript has as main objective to prove that the addition of ceria nanoparticles decreases the overpotential towards water oxidation in acidic media while maintaining the long term (100h) stability of the composite material. In general the manuscript is very well written and the data support the conclusions drawn by the authors. The question now is if the reported results are important enough to be published in Nature Communications. As a non expert in electrochemistry I leave the answer to the other reviewers and the editor, on the other hand I am not convinced that a decrease of overpotential of ca. 50mV with the addition of ceria nanoparticles under otherwise identical conditions would deserve publication in a very high impact journal. If the role of ceria and the mechanism could be more deeply elucidated and discussed, that would be certainly the case. Especially, as far as I understood the main conclusion stating that "the electronic modulation between Co₃O₄ and CeO₂ creates a more favorable local bonding environment without altering the crystal structure and suppresses the charge accumulation of Co₃O₄ under electrochemical conditions, which are the keys to bypassing the potential-determining redox step in Co₃O₄ and thus enhancing the acidic OER activity." is not fully supported and demonstrated by the study at this stage.

Reviewer #3 (Remarks to the Author):

Huang et al. report a study of Co₃O₄/CeO₂ nanocomposite as electrocatalyst for the oxygen evolution reaction (OER) in acidic medium. The choice of these materials is not new, the new aspect – repeatedly stressed by the authors – is the application under acidic conditions. It is also surprising that ceria is assumed to be “catalytically inert”, even more so as nanocrystalline CeO₂ is used. It is in fact well documented in thermal catalysis that mixed oxides of Co₃O₄ and (the much less active) CeO₂ exhibit synergistic catalytic effects, due to modified redox properties and easier oxygen vacancy formation energy. Thus, this is expected to also affect electrocatalysis.

The authors have carried out a very extensive electrochemical study, unfortunately complemented by solely pre- and post-reaction analysis. It is a pity that no in situ characterization was performed, as it was previously demonstrated that such active surfaces do restructure (including surface compositional changes) during electrochemical reactions, which is why in situ or preferentially operando characterization (e.g. spectroscopy simultaneous with electrochemistry) would be desirable. An active electrocatalyst is reported but at this state – and for the reasons given below – an in-depth atomic scale explanation of the activity enhancement is still lacking and remains rather speculative.

Comments:

1. As the crystal structure of modified Co₃O₄ remains unaltered, this suggests that mainly the surface of Co₃O₄ is involved, which contradicts a CeO₂ “incorporation” (in the Co₃O₄ lattice? the term was not really defined). For surface-governed processes, structure characterization by SEM and PXRD is rather insufficient.

2. The HRTEM and EDX characterization seems contradictory. Whereas for the nanocomposite the first shows separate nanocrystals of Co₃O₄ and CeO₂, the latter indicates a quite homogeneous distribution of Co and Ce. Does the surface Ce metal content of ~8% match the abundance of CeO₂ crystals in HRTEM? The morphology of Co₃O₄ and Co₃O₄/CeO₂ is quite different so that terminations may strongly vary as well.

3. The spinel oxide Co₃O₄ and cubic CeO₂ structures may not form macroscopic mixed solutions, but incorporation of Ce atoms in the Co₃O₄ surface (and vice versa?) is possible and would be undetected by any of the applied methods. This would explain why the redox properties of Co₃O₄ were unaffected by the degree of crystallinity and why the OER activity of Co₃O₄/CeO₂ remained nearly constant regardless of the different crystallinity. Such Ce surface species may leach during reaction, explaining the lower Ce content in post-reaction XPS and EDX.

4. The ex situ XANES and EXAFS measurements are not convincing as they are on the one hand not surface sensitive (and average over all crystals) and on the other hand can not explain the “incorporation”. If Ce is incorporated, but still has only a marginal effect on the coordination of the Co ions inside the crystals (“subtle changes; slightly higher Co oxidation state”), how can this affect the active sites on the crystal surface?

5. The central topic of “electronic modulation that regulates the redox properties” would need to be substantiated. At this point explanations remain rather handwaving: “enhanced acidic OER activity is more likely to be caused by the change in the local bonding environment of Co₃O₄ induced by CeO₂” and “rather enhances the intrinsic activity of the same type of catalytic active site in Co₃O₄ by modifying the entropy of activation and the concentration of active sites”. How can this be envisioned in an atomic picture? What about interface effects between the grains, also involving a CoCe redox couple?

It is acknowledged that the authors have reported a highly active earth-abundant OER electrocatalyst in acidic medium, with thorough electrochemical characterization. However, the true origin of this effect remains largely unknown. In addition to in situ and/or operando techniques (that also have to cope with the typically minute number of active sites), theoretical studies may be crucial to solve this puzzle.

Response Letter

We thank all three Reviewers for their positive reviews of this manuscripts and the constructive suggestions that help to improve the scientific presentation of this manuscript. Point-by-Point responses to address the concerns raised by the three Reviewers are shown in the following.

Reviewer #1 (Remarks to the Author):

This is an interesting manuscript on the enhancement on the activity of Co oxide as a water oxidation electrocatalyst in acidic media, when decorated with ceria. The authors performed multiple and complementary analyses, including consistent characterization and mechanistic studies, resulting in a very interesting manuscript that I think really deserves to be published in Nat Commun.

We appreciate the Reviewer for the positive comments on our manuscript.

I have two important comments, though, that I hope the authors can address in a revised version:

1) The authors compare the activity of Co₃O₄/CeO₂ vs Co₃O₄ in their conditions. However, they do not compare their results in detail with previous reports on Co oxides as OER electrocatalyst in acidic media. I think it would help if the authors could comment on previous, important references such as J. Am. Chem. Soc, 136(8), 3304–3311 and some others to put their results into the proper context.

Response: Thanks for the comments. For better comparison in context, we have summarized the important OER electrocatalysts and compared our catalysts to them in Supplementary Table 3. We have also cited this relevant literature as Ref. 70 and made the comparison based on the new metal dissolution results at open circuit potential. Please see the **Response to next comment** for further details.

2) The authors are quite honest stating that this OER enhancement does not improve Co oxide stability in acid. However, I miss some explicit experiments to show the stability of the catalysts in acid at open circuit potential. If I understood it right, the authors explore the stability during OER, at 10 mA cm⁻². In these conditions, Co₃O₄ is much more stable than without a bias. This open circuit instability is also crucial.

Response: We thank the Reviewer for the great suggestion. We have further carried out the stability measurement on the catalysts at open circuit potential accordingly. The dissolution of both samples without a bias was monitored by inductively coupled plasma optical emission spectroscopy (ICP-OES), as shown in the Figure R1 below (added into SI as the new Supplementary Fig. 30). At open circuit potential, the Co₃O₄/CeO₂ composite catalyst showed better stability compared to Co₃O₄ with slower Co dissolution, with no obvious dissolution of Ce

even after 10 h, suggesting the CeO_2 is stable without bias in 0.5 M H_2SO_4 solution, which might protect the Co_3O_4 from corrosion. We have proved that both Co_3O_4 and $\text{Co}_3\text{O}_4/\text{CeO}_2$ showed similar dissolution rate of Co under OER condition (Supplementary Fig. 29b). However, both samples are less stable without a bias, since after the same duration of 4 hours, the leached Co concentration is an order of magnitude higher than that with a bias (Figure R1b). Thus the applied potential can enhance the stability of Co_3O_4 . To address the stability differences with or without a bias, we have added more discussion in the text in page 14 and cited the related reference.

Figure R1. The metal dissolution rates of Co_3O_4 and $\text{Co}_3\text{O}_4/\text{CeO}_2$ catalysts without a bias. b) The comparison of the amount of Co ions in the electrolyte solutions leached from both Co_3O_4 and $\text{Co}_3\text{O}_4/\text{CeO}_2$ catalysts after 4 h with or without a bias.

Reviewer #2 (Remarks to the Author):

The manuscript by Huang et al. reports the effect of the addition of ceria nanocrystals into a nanostructured cobalt oxide electrocatalyst for the electrochemical water oxidation in 0.5 M sulfuric acid aqueous solutions. The material science part is relatively limited, the material is electrodeposited onto fluorine doped tin dioxide and characterized after calcination by powder x-ray diffraction and electron microscopy, giving only a superficial picture of the chemical and structural properties of the film. Especially, the high resolution electron micrographs are of relatively low quality (Figure 1c,d) and their analysis not sufficient (interplanar distances are measured in only one direction for each crystallite and large zones are poorly colored masking the real information of the image). The same applies for the EDX maps, which present a too low resolution to distinguish the individual Ce and Co-containing crystallites. From the data presented the most performing material is a mixture of sub-10 nm nanocrystals of Co_3O_4 and CeO_2 with a 10 weight% content.

Response: Thanks for the suggestions! We have further collected better PXRD and high-resolution TEM data to strengthen the structural characterization and the understanding on the catalysts. The major additions include:

I). We have updated the PXRD pattern for $\text{Co}_3\text{O}_4/\text{CeO}_2$ (Figure R2 / updated Fig. 1e in the revised manuscript), which further demonstrates the composite nature of the catalyst. The Scherrer analysis reveals the average crystal domain size at ~ 5 nm for CeO_2 (Figure R2b / new Supplementary Fig. 6).

Figure R2. (a) PXRD patterns of the Co_3O_4 and $\text{Co}_3\text{O}_4/\text{CeO}_2$ on FTO substrates in comparison with the standard PXRD pattern of Co_3O_4 (JCPDS 43-1003) and CeO_2 (JCPDS 43-1002). (b) The enlarged PXRD pattern of $\text{Co}_3\text{O}_4/\text{CeO}_2$ at around the CeO_2 (111) peak position. Scherrer analysis of this peak at a full width at half maximum of ~ 1.6 degree reveals the average crystal domain size of CeO_2 is around ~ 5 nm.

II). The improved high-resolution TEM images are shown in Figure R3 below and in updated Figure 1c-d in the revised manuscript. Following Reviewer's suggestion, we have also removed the colored masking. The interplanar distance was also measured along two crystallographic

directions to improve the accuracy. The CeO_2 domain in the $\text{Co}_3\text{O}_4/\text{CeO}_2$ composite is denoted with a yellow dashed circle, with a crystal size of ~ 5 nm, matching well with the estimate from the PXRD pattern (Figure R2b). The crystal size of CeO_2 is very small, and CeO_2 and Co_3O_4 are highly mixed in $\text{Co}_3\text{O}_4/\text{CeO}_2$ due to the co-deposition method.

Figure R3. HRTEM images of (a) Co_3O_4 and (b) $\text{Co}_3\text{O}_4/\text{CeO}_2$ nanosheets.

We apologize for not getting better element mapping images with higher quality due to the limitation of the TEM instrument. But to further support the structure characterization of these samples, we have added a new Supplementary Fig. 4 with more HRTEM images in the SI, also shown as Figure R4 on the next page. To better illustrate the $\text{Co}_3\text{O}_4/\text{CeO}_2$ sample, the distribution of CeO_2 crystallites in the composite is schematically shown in Figure R4e.

Figure R4. Additional HRTEM images for (a-b) Co_3O_4 and (c-d) $\text{Co}_3\text{O}_4/\text{CeO}_2$ catalysts. (b) and (d) are the enlarged images from the regions in (a) and (c) highlighted by dashed boxes, respectively. (e) The schematic illustration for the distribution of CeO_2 domains in the $\text{Co}_3\text{O}_4/\text{CeO}_2$ composite.

On the other hand, the electrochemical part is very well detailed and probably carried out following state of the art procedures (not being an expert, I can not properly judge the electrochemical and largest part of the manuscript). The manuscript has as main objective to prove that the addition of ceria nanoparticles decreases the overpotential towards water oxidation in acidic media while maintaining the long term (100h) stability of the composite material. In general the manuscript is very well written and the data support the conclusions drawn by the authors. The questions now is if the reported results are important enough to be published in Nature Communications. As a non expert in electrochemistry I leave the answer to the other reviewers and the editor, on the other hand I am not convinced that a decrease of overpotential of ca. 50mV with the addition of ceria nanoparticles under otherwise identical conditions would deserve publication in a very high impact journal.

Response: Thank you for the approval of the electrochemical characterizations in this manuscript. The overpotential improvement at the geometric current density of 10 mA cm^{-2} (η_{10}) for the $\text{Co}_3\text{O}_4/\text{CeO}_2$ composite catalyst on FTO is around $\sim 84 \text{ mV}$, which is a significant advance for acidic oxygen evolution reaction (OER), especially considering non-noble metals are used. In fact, the catalytic performance can be further improved by changing FTO to a porous substrate such as carbon paper. In this case, the η_{10} of our $\text{Co}_3\text{O}_4/\text{CeO}_2$ composite is 347 mV and only 46 mV higher than the benchmark RuO_2 catalyst for acidic OER, suggesting it is one of the best Earth-abundant catalysts for acidic OER (see a comparison in Supplementary Table 3).

If the role of ceria and the mechanism could be more deeply elucidated and discussed, that would be certainly the case. Especially, as far as I understood the main conclusion stating that “the electronic modulation between Co_3O_4 and CeO_2 creates a more favorable local bonding environment without altering the crystal structure and suppresses the charge accumulation of Co_3O_4 under electrochemical conditions, which are the keys to bypassing the potential-determining redox step in Co_3O_4 and thus enhancing the acidic OER activity.” is not fully supported and demonstrated by the study at this stage.

Response: Indeed the understanding of the mechanism for the catalytic enhancement due to the inclusion of CeO_2 in $\text{Co}_3\text{O}_4/\text{CeO}_2$ composite is a highlight of this work. Because such discussion invokes conceptual discussion in diverse topics of electrochemistry, electrocatalysis, electronic structures and nanomaterials, we further explain and clarify these points below:

I). The role of ceria.

We have conclusively shown that CeO_2 is not active for acidic OER compared to the Co_3O_4 (Figure R5a below / new Supplementary Fig. 9b). OER is a four-electron process coupled with proton. To investigate the impact on proton transfer, we have performed H/D isotope kinetic effect (KIE) experiments on the two catalysts to reveal similar KIE (Figure R5b). Therefore, CeO_2 does not change the proton transfer properties during OER. The X-ray adsorption spectroscopy (XAS) data (Figure 4a-c) already revealed that the modified bonding environments around Co active sites and the more flexible local bonding make $\text{Co}_3\text{O}_4/\text{CeO}_2$ more active. ***These understandings will be further supported by the newly added in situ Raman data discussed below.***

Figure R5. (a) The CV curve of the CeO_2 sample compared to that of Co_3O_4 on carbon paper in

0.5 M H₂SO₄ solution. Inset is enlarged image to see the redox features. Clearly, the CeO₂ sample shows no obvious redox features and poor OER activity in acid compared to Co₃O₄. (b) The KIE curves plotted with the LSV curves on the overpotential scale.

II). The reaction mechanism.

The introduction of CeO₂ does not change the active-sites. The similar active-sites in Co₃O₄/CeO₂ is confirmed by the activation energy analysis (presented in Figure 3f). Based on the redox features of Co₃O₄ and previous literature, we have proposed the active-site evolution during the OER reaction as shown in Figure R6 (Supplementary Fig. 7).

Figure R6. Proposed structural motifs associated with the three sets of pre-OER redox features present in the Co₃O₄ catalyst in acidic media that involve dimeric Co redox centers.

III). New *in situ* Raman data to support the conclusion.

To further unveil the relationships between the electronic modulation, redox-mediated surface reconstruction, and catalytic activity of Co₃O₄/CeO₂ vs. Co₃O₄, we have further performed *in situ* Raman experiments on both catalysts in 0.5 M H₂SO₄ solution under OER conditions. We summarize the key finding below.

Both dry samples of Co₃O₄ and Co₃O₄/CeO₂ display four characteristic Raman peaks corresponding to the E_g (~ 480 cm⁻¹), F_{2g} (~ 520 cm⁻¹), F_{2g} (~ 620 cm⁻¹) and A_{1g} (~ 690 cm⁻¹) phonon modes of Co₃O₄ spinel oxides (Figure R7a)¹. After the samples were immersed in the 0.5 M H₂SO₄ electrolyte, a new Raman peak emerged at ~ 600 cm⁻¹ at the applied potential of 1.22 V vs. RHE, which is attributed to the formation of CoOOH species at the surface². This CoOOH species was less clearly detected at high potentials and started to disappear from the Co₃O₄/CeO₂ and Co₃O₄ surfaces at 1.52 V and 1.62 V vs. RHE, respectively, which coincided with their respective OER onset potentials (Supplementary Fig. 25), as well as the two pre-OER redox features of Co₃O₄ associated with Co^{III}Co^{III} ↔ Co^{III}Co^{IV} (~ 1.50 V vs. RHE) and Co^{III}Co^{IV} ↔ Co^{IV}Co^{IV} (~ 1.63 V vs. RHE) transitions (Fig. 2b). Clearly this CoOOH species is not the actual active phase for acidic OER and needs to be further oxidized into Co^{IV} species. The disappearance of this CoOOH species from Co₃O₄/CeO₂ at a lower potential, indicating that it is easier to oxidize the active Co sites in the Co₃O₄/CeO₂ catalyst into OER-active Co^{IV} species compared to those Co sites in the pure Co₃O₄, due to the electronic modulation effect of CeO₂. The intensities of all Raman peaks at higher applied potentials decrease significantly (Figure R7b and lower panel in Figure R7c), which was usually accompanied with the increase in average valence state of Co atoms³. When the applied potential was finally switched back from 1.87 V to 1.22 V vs. RHE, the peak intensities partially recovered (lower panel in Figure R7c) and the CoOOH species was clearly detected again.

Figure R7. *In situ* Raman characterizations of Co_3O_4 and $\text{Co}_3\text{O}_4/\text{CeO}_2$ catalysts on carbon paper electrodes during OER testing in 0.5 M H_2SO_4 solution to reveal the structural evolution of catalysts. (a) The *in situ* Raman spectra of Co_3O_4 (left panel) and $\text{Co}_3\text{O}_4/\text{CeO}_2$ (right panel) at various constant potentials (vs. RHE) without iR correction (increased from 1.22 V to 1.87 V and then back to 1.22 V). The Raman spectra of the dry samples were also presented at the bottom for comparisons. (b) The Raman A_{1g} peaks of Co_3O_4 (top) and $\text{Co}_3\text{O}_4/\text{CeO}_2$ (bottom) were fitted with Lorentzian function to extract the peak positions, intensity, and FWHM (dash lines: raw spectra; dots: fitting results). (c) The Raman A_{1g} peak positions (upper panel) and intensity ratios with respect to the initial intensity at 1.22 V (lower panel) plotted against the applied potential. The open symbols represent the data collected at 1.22 V at the end after applying the higher potential sequence.

To understand the evolution of the local bonding environment at the catalyst surface during the OER process, the peak position, intensity, and full width at half maximum (FWHM) of the Raman A_{1g} peak ($\sim 690 \text{ cm}^{-1}$) were extracted by fitting with Lorentzian function (Figure R7b,c). The shift in the peak position as a function of applied potential can be interpreted as either the change in crystalline domain size or the generation of surface strain/stress (i.e. lattice contraction/extension)^{4,5}. Since the marginal variations in the peak FWHM suggest the crystalline domain sizes of both samples remain relatively constant during the OER process (Figure R8, new supplementary Fig. 26), the observed peak position shift should result from the lattice contraction/extension and the surface reconstruction due to the changing local bonding environment. More importantly, the peak positions shift in opposite directions on these two catalysts as the potential goes over the OER catalytic onsets (Figure R7c upper panel). $\text{Co}_3\text{O}_4/\text{CeO}_2$ shows a red-shift in the A_{1g} peak position after the onset of OER at 1.52 V vs. RHE. Red-shifts in Raman signals are commonly observed in OER catalysts (CoO_x ^{3,6}, NiOOH ⁷, NiFe and CoFe oxyhydroxides⁸) at OER operating potentials, and they generally reflect the characteristic vibration for local bonding environment at the outer layer of catalysts with oxidized active site during OER. Thus the generation of active Co^{IV} species that can participate in a fast and

efficient OER process should lead to the observed red-shift of the Raman signals. In contrast, blue-shifts in Raman signals usually suggest lattice contraction and charge redistribution^{4,9}. Unlike the more active $\text{Co}_3\text{O}_4/\text{CeO}_2$, the pure Co_3O_4 catalyst would go through a significant charge-accumulation surface reconstruction ($\text{Co}^{\text{III}}\text{Co}^{\text{IV}} \leftrightarrow \text{Co}^{\text{IV}}\text{Co}^{\text{IV}}$) at ~ 1.62 V around the onset for OER. The Co^{IV} species generated during this process are stabilized and cannot participate in fast OER turnover since the reduction peak could be still observed when the potential was scanned backwards, thus they lead to a blue-shift in the Raman signals (Figure R7c). Another interesting difference is that the peak position of $\text{Co}_3\text{O}_4/\text{CeO}_2$ at 1.22 V vs. RHE remains almost unchanged before and after applying the higher potential sequence, suggesting the flexibility in the local bonding environment of Co_3O_4 in the composite catalyst. However, the peak position of Co_3O_4 cannot fully recover after the same potential cycle, with the final peak at ~ 1 cm^{-1} higher in wavenumber (Figure R9), which is consistent with the positive charge accumulated at the Co center with shorter Co-O bond in the Co_3O_4 sample after OER (Fig. 4a-c). Together with the *ex situ* XAS results, the *in situ* Raman results clearly demonstrate that the bonding environments surrounding the Co centers are affected by the electronic modulation between Co_3O_4 and CeO_2 across the interface in the nanocomposite $\text{Co}_3\text{O}_4/\text{CeO}_2$ catalyst, which allows the Co active sites to be more readily oxidized and avoid the significant potential-determining surface reconstruction (forming dimeric $\text{Co}^{\text{IV}}\text{Co}^{\text{IV}}$) with charge accumulation and lattice contraction. As Co^{IV} is the key intermediate to start OER process, the more facile formation of Co^{IV} species and de-stabilization of $\text{Co}^{\text{IV}}\text{Co}^{\text{IV}}$ in $\text{Co}_3\text{O}_4/\text{CeO}_2$ would allow faster OER kinetics thus enhance the catalytic activity.

Figure R8. The FWHM of the A_{1g} peaks from the *in situ* Raman spectra for Co_3O_4 and $\text{Co}_3\text{O}_4/\text{CeO}_2$ plotted against the applied potential show relatively small variations. Larger FWHM suggests smaller crystallites in $\text{Co}_3\text{O}_4/\text{CeO}_2$, which is consistent with the XRD and HRTEM results.

Figure R9. The Raman spectra for (a) Co_3O_4 and (b) $\text{Co}_3\text{O}_4/\text{CeO}_2$ at the 1.22 V vs. RHE before and after OER.

The Figure R7 has been added as a new Figure 5 in the revised manuscript and significant discussion about the insights gained from these *in situ* Raman experiments surrounding this new Figure 5 has been added to page 11-13 of the revised manuscript.

Reviewer #3 (Remarks to the Author):

Huang et al. report a study of $\text{Co}_3\text{O}_4/\text{CeO}_2$ nanocomposite as electrocatalyst for the oxygen evolution reaction (OER) in acidic medium. The choice of these materials is not new, the new aspect – repeatedly stressed by the authors – is the application under acidic conditions. It is also surprising that ceria is assumed to be “catalytically inert”, even more so as nanocrystalline CeO_2 is used. It is in fact well documented in thermal catalysis that mixed oxides of Co_3O_4 and (the much less active) CeO_2 exhibit synergistic catalytic effects, due to modified redox properties and easier oxygen vacancy formation energy. Thus, this is expected to also affect electrocatalysis.

*The authors have carried out a very extensive electrochemical study, unfortunately complemented by solely pre- and post-reaction analysis. It is a pity that no *in situ* characterization was performed, as it was previously demonstrated that such active surfaces do restructure (including surface compositional changes) during electrochemical reactions, which is why *in situ* or preferentially operando characterization (e.g. spectroscopy simultaneous with electrochemistry) would be desirable. An active electrocatalyst is reported but at this state – and for the reasons given below – an in-depth atomic scale explanation of the activity enhancement is still lacking and remains rather speculative.*

Response: We thank the Reviewer for confirming the aspects of our findings that are new and for approving our extensive electrochemical studies, and for making many insightful comments. We address the many points brought up in this summary in details below, or in the specific comments later.

It is also surprising that ceria is assumed to be “catalytically inert”. We have tested the catalytic behaviors of CeO₂ in 0.5 M H₂SO₄ solution (Figure R5a shown below / updated Supplementary Fig. 9b) and confirmed that it shows very poor OER activity compared to Co₃O₄. We agree with the Reviewer that CeO₂ exhibits synergistic catalytic effect, but CeO₂ itself clearly is not the active center or active phase for electrocatalytic OER. But we realize that the phrase “catalytically inert” might be possibly interpreted as no synergistic effect. To avoid misleading the readers, we have addressed CeO₂ is not active center but deleted the “catalytically inert” in the text throughout the manuscript.

Figure R5a. The CV curve of the CeO₂ sample compared to that of Co₃O₄ on carbon paper in 0.5 M H₂SO₄ solution. Inset is the enlarged image to see the redox features. Clearly, the CeO₂ sample shows no obvious redox features and poor OER activity in acid compared to Co₃O₄.

It is in fact well documented in thermal catalysis that mixed oxides of Co₃O₄ and (the much less active) CeO₂ exhibit synergistic catalytic effects... We thank the Reviewer for bringing this to our attention! Previously, we have discussed the applications of CeO₂ in electrocatalysis but missed the literature in thermal catalysis. We have added more discussions about the applications of such nanocomposites in thermal catalysis in the Introduction on page 3 and cited a related review paper (ref. 37).

*It is a pity that no in situ characterization was performed ... We have further carried out in situ Raman characterizations to address the concerns raised by the Reviewer, **which will be discussed in details later**. Unfortunately, in situ or operando synchrotron X-ray spectroscopic studies have not been possible to schedule in a short timeframe, especially during the pandemic when no outside users have been allowed to perform experiments at the APS beamlines where our ex situ X-ray spectroscopic results were collected.*

Comments:

1. As the crystal structure of modified Co₃O₄ remains unaltered, this suggests that mainly the surface of Co₃O₄ is involved, which contradicts a CeO₂ “incorporation” (in the Co₃O₄ lattice? the term was not really defined). For surface-governed processes, structure characterization by SEM and PXRD is rather insufficient.

Response: We appreciate the valuable comment, but realize in hindsight that there was misunderstanding about how these $\text{Co}_3\text{O}_4/\text{CeO}_2$ catalysts look like by Reviewer 3, and this misconception likely systematically impacted how our discussion was interpreted by Reviewer 3 that caused most of the questions raised here. We would like to clarify that the catalysts here are indeed **nanocomposites of nanocrystalline domains of both Co_3O_4 and CeO_2** interdispersed with each other, these two compounds are **NOT making alloyed solutions**, and the **Ce are not just on the surface of Co_3O_4 domains**. Perhaps our poor choice of words and lack of the elaboration in the original text caused this misunderstanding. Below in responding to each questions, we will present various structural characterization results, including several key results that have been added or improved during revision, to support this correct structural interpretation, which we believe is consistent with all data.

The clearest evidences for the nanocomposite nature of the $\text{Co}_3\text{O}_4/\text{CeO}_2$ catalysts are the high-resolution TEM images, which we have updated and enhanced. We have replaced the original high resolution TEM images in Figure 1c-d with more clear ones without color masking (reproduced below as Figure R3). We have added a new Supplementary Fig. 4 with more HRTEM images in the SI, also shown as Figure R4 on the next page. To better illustrate the $\text{Co}_3\text{O}_4/\text{CeO}_2$ sample, the distribution of CeO_2 crystallites in the composite is schematically shown in Figure R4e. These images show that the CeO_2 crystallites of about 5 nm in size are inlaid with Co_3O_4 rather than only dispersed at the surface. The CeO_2 crystalline domain size of ~ 5 nm also matches the estimate from the Scherrer analysis of PXRD peak (new Supplementary Fig. 6, shown earlier as Figure R2 in a response to Reviewer #2). To more accurately describe these nanocomposite structures, the word “incorporation” has been replaced with “introduction” in the manuscript text, which has been better defined with more elaboration to avoid misleading the readers.

Figure R3. HRTEM images of (a) Co_3O_4 and (b) $\text{Co}_3\text{O}_4/\text{CeO}_2$ nanosheets.

Figure R4. Additional HRTEM images for (a-b) Co_3O_4 and (c-d) $\text{Co}_3\text{O}_4/\text{CeO}_2$ catalysts. (b) and (d) are the enlarged images from the regions in (a) and (c) highlighted by dashed boxes, respectively. (e) The schematic illustration for the distribution of CeO_2 domains in the $\text{Co}_3\text{O}_4/\text{CeO}_2$ composite.

But we do agree with the Reviewer that it is important to investigate the surface of the catalysts. Actually we had conducted the *surface-sensitive X-ray photoelectron spectroscopy* (XPS) measurements previously (now the Supplementary Fig. 22). No obvious alteration has been observed in both binding energy and valence state of the catalysts after the OER test. During revision, we have further conducted *in situ Raman experiments*. The Raman spectra collected at 1.22 V vs. RHE before and after increasing the potential to 1.87 V vs. RHE at the OER region showed obvious blue shifts ($\sim 1 \text{ cm}^{-1}$) for all the peaks in Co_3O_4 (Figure R9a), which is usually caused by the charge redistribution and compression in the lattice.^{4,5} This is consistent with the XAS results (Fig. 4a-c) which show obvious change in the average valence state and contraction in the Co-O bond length. The changes in Raman and XAS spectra for $\text{Co}_3\text{O}_4/\text{CeO}_2$ are much less significant. These new observations are consistent with the trends we observed by the XAS characterizations. These Raman and XAS characterizations further build the connections of the

local bonding environments at the surface and bulk, thus lead to a more conclusive understanding on the structure properties of the catalysts. This Figure R6 has been added as the new Supplementary Fig. 27.

Figure R9. The Raman spectra for (a) Co_3O_4 and (b) $\text{Co}_3\text{O}_4/\text{CeO}_2$ at the 1.22 V vs. RHE before and after OER.

2. The HRTEM and EDX characterization seems contradictory. Whereas for the nanocomposite the first shows separate nanocrystals of Co_3O_4 and CeO_2 , the latter indicates a quite homogeneous distribution of Co and Ce. Does the surface Ce metal content of ~8% match the abundance of CeO_2 crystals in HRTEM? The morphology of Co_3O_4 and $\text{Co}_3\text{O}_4/\text{CeO}_2$ is quite different so that terminations may strongly vary as well.

Response: Sorry for the confusion here. The interpretation by the Reviewer from HRTEM about “separate nanocrystals of Co_3O_4 and CeO_2 ” is correct (see more images on last two pages), the reason that we could not resolve the individual CeO_2 crystallite in the composite from elemental mapping is due to the poor spatial resolution of this EDX mapping. This is partially due to the limitation of the TEM instrument. The other reason is that the detection depth for the EDX technique is beyond the thickness of the nanosheet samples here (~ 10 to 20 nm), so the elemental mapping would collect all the signals throughout the disordered composite, which explains a relatively homogeneous dispersion of Ce in the elemental mapping. The percentage of Ce from EDS (~ 9.1%) is generally consistent with the CeO_2 domains observed by the HRTEM in Figure R4c (which can be estimated from the area of the CeO_2 domains among all the crystallites).

Co_3O_4 and $\text{Co}_3\text{O}_4/\text{CeO}_2$ display similar nanosheet-like morphology with similar polycrystallinity feature and Co_3O_4 crystal domain size of 13.9 and 9.7 nm, respectively. The diffraction fringes for planes such as (220), (111) and (311) are generally observed in both samples, so we do not believe there is significant difference in crystal termination.

3. The spinel oxide Co_3O_4 and cubic CeO_2 structures may not form macroscopic mixed solutions, but incorporation of Ce atoms in the Co_3O_4 surface (and vice versa?) is possible and would be

undetected by any of the applied methods. This would explain why the redox properties of Co_3O_4 were unaffected by the degree of crystallinity and why the OER activity of $\text{Co}_3\text{O}_4/\text{CeO}_2$ remained nearly constant regardless of the different crystallinity. Such Ce surface species may leach during reaction, explaining the lower Ce content in post-reaction XPS and EDX.

Response: We thank the Reviewer for providing this hypothesis to explain the unique redox properties for the Co_3O_4 and $\text{Co}_3\text{O}_4/\text{CeO}_2$. We agree that the Co_3O_4 and CeO_2 should not form mixed alloyed solutions. We agree that there was no straightforward characterization that can exclude the possible existence of Ce atoms on the surface of Co_3O_4 previously, but we do not agree that such a possibility can be deduced according to the metal dissolution behaviors during OER. To verify the hypothesis from the Reviewer, we have further conducted the dissolution experiment at the open circuit potential (Figure R1, shown below). However, the Co dissolution rate was much faster than that of Ce at the open circuit potential (Figure R1a); in fact, no obvious Ce dissolution was detected at the open circuit voltage. This is inconsistent with the hypothesis that Ce atoms are enriched on the surface of Co_3O_4 . To conclude, the $\text{Co}_3\text{O}_4/\text{CeO}_2$ catalyst presented here is composite in nature based on these new leaching results at open circuit potential and extensive structural characterization.

Figure R1. The metal dissolution rates of Co_3O_4 and $\text{Co}_3\text{O}_4/\text{CeO}_2$ catalysts without a bias. b) The comparison of the amount of Co ions in the electrolyte solutions leached from both Co_3O_4 and $\text{Co}_3\text{O}_4/\text{CeO}_2$ catalysts after 4 h with or without a bias.

4. The *ex situ* XANES and EXAFS measurements are not convincing as they are on the one hand not surface sensitive (and average over all crystals) and on the other hand can not explain the “incorporation”. If Ce is incorporated, but still has only a marginal effect on the coordination of the Co ions inside the crystals (“subtle changes; slightly higher Co oxidation state”), how can this affect the active sites on the crystal surface?

Response: Thanks for the comment. We totally agree with the Reviewer that: 1) the XAS characterizations are probing the bulk rather than surface sensitive; 2) only the outer layer of Co is participating during OER. The confusion here arises from the general misunderstanding of the catalyst here. Now we have proven the composite nature of $\text{Co}_3\text{O}_4/\text{CeO}_2$ in Figure R1 and established the local domain features in Figure R4 (see Responses to Question 1 above): the very

small nanocrystalline domains of CeO₂ (~ 5 nm) and Co₃O₄ (~ 9.7 nm) are interdispersed in the Co₃O₄/CeO₂ nanocomposite with numerous contact regions between Co₃O₄ and CeO₂, thus enhancing the modulation effects due to the introduction of the CeO₂ domains. The resulting structural changes are “global” and throughout all of the nanocrystalline domains of Co₃O₄, which we experimentally observed with the XANES and EXAFS measurements (averaged over all domains). These results are truly reflecting the changes in structure properties before and after OER, including those Co active sites on the surfaces which are more consequential for the catalytic processes. Now we have further conducted *in situ* Raman characterizations as will be discussed in details below in the responses to Question 5. The *in situ* Raman and *ex situ* XAS characterizations have consistently proven the modifications in local bonding environments in the Co₃O₄/CeO₂ due to the introduction of CeO₂ and help us to understand the origin of the catalytic enhancement.

5. The central topic of “electronic modulation that regulates the redox properties” would need to be substantiated. At this point explanations remain rather handwaving: “enhanced acidic OER activity is more likely to be caused by the change in the local bonding environment of Co₃O₄ induced by CeO₂” and “rather enhances the intrinsic activity of the same type of catalytic active site in Co₃O₄ by modifying the entropy of activation and the concentration of active sites”. How can this be envisioned in an atomic picture? What about interface effects between the grains, also involving a CoCe redox couple?

Response: I) We agree with the Reviewer that *in situ* characterizations is important to directly connect between the electronic modulation, redox properties (surface reconstruction), and catalytic activity. This would provide more support to our central mechanistic explanation of the observed catalytic enhancements. Therefore, we have designed and performed *in situ* Raman experiments on these catalysts during OER reactions. Figure R10 (new Supplementary Fig. 25) shows the technical details of the *in situ* Raman experimental setup and electrochemical cycling procedures for the experiments. The Raman spectra have been collected at constant potentials without *iR* correction from 1.22 to 1.87 V then back to 1.22 V (all vs. RHE) again. The Co₃O₄ catalyst shows characteristic Co^{III}Co^{IV} ↔ Co^{IV}Co^{IV} redox and poorer OER performance compared to Co₃O₄/CeO₂ (Figure R10b). The obvious OER current density detected by chronoamperometry technique is at 1.62 V and 1.52 V vs. RHE for Co₃O₄ and Co₃O₄/CeO₂, respectively, suggesting the onset of OER in Co₃O₄/CeO₂ is ~ 100 mV lower than the Co₃O₄.

Figure R10. (a) The experimental setup for *in situ* Raman measurements. (b) The CV curves of Co₃O₄ and Co₃O₄/CeO₂ catalysts on carbon paper electrodes collected at the scan rate of 20 mV s⁻¹ in 0.5 M H₂SO₄ using the *in situ* Raman cell. (c,d) The chronoamperometry curves of (c) Co₃O₄ and (d) Co₃O₄/CeO₂ collected at various constant potentials (vs. RHE) during the *in situ* Raman measurements.

Both dry samples of Co₃O₄ and Co₃O₄/CeO₂ display four characteristic Raman peaks corresponding to the E_g (~ 480 cm⁻¹), F_{2g} (~ 520 cm⁻¹), F_{2g} (~ 620 cm⁻¹) and A_{1g} (~ 690 cm⁻¹) phonon modes of Co₃O₄ spinel oxides (Figure R7a)¹. After the samples were immersed in the 0.5 M H₂SO₄ electrolyte, a new Raman peak emerged at ~ 600 cm⁻¹ at the applied potential of 1.22 V vs. RHE, which is attributed to the formation of CoOOH species at the surface². This CoOOH species was less clearly detected at high potentials and started to disappear from the Co₃O₄/CeO₂ and Co₃O₄ surfaces at 1.52 V and 1.62 V vs. RHE, respectively, which coincided with their respective OER onset potentials (Supplementary Fig. 25), as well as the two pre-OER redox features of Co₃O₄ associated with Co^{III}Co^{III} ↔ Co^{III}Co^{IV} (~ 1.50 V vs. RHE) and Co^{III}Co^{IV} ↔ Co^{IV}Co^{IV} (~ 1.63 V vs. RHE) transitions (Fig. 2b). Clearly this CoOOH species is not the actual active phase for acidic OER and needs to be further oxidized into Co^{IV} species. The disappearance of this CoOOH species from Co₃O₄/CeO₂ at a lower potential indicates that it is easier to oxidize the active Co sites in the Co₃O₄/CeO₂ catalyst into OER-active Co^{IV} species compared to those Co sites in the pure Co₃O₄, due to the electronic modulation effect of CeO₂. The intensities of all Raman peaks at higher applied potentials decrease significantly (Figure R7b and lower panel in Figure R7c), which was usually accompanied with the increase in average valence state of Co atoms³. When the applied potential was finally switched back from 1.87 V to 1.22 V vs. RHE, the

peak intensities partially recovered (lower panel in Figure R7c) and the CoOOH species was clearly detected again.

Figure R7. *In situ* Raman characterizations of Co₃O₄ and Co₃O₄/CeO₂ catalysts on carbon paper electrodes during OER testing in 0.5 M H₂SO₄ solution to reveal the structural evolution of catalysts. (a) The *in situ* Raman spectra of Co₃O₄ (left panel) and Co₃O₄/CeO₂ (right panel) at various constant potentials (vs. RHE) without *iR* correction (increased from 1.22 V to 1.87 V and then back to 1.22 V). The Raman spectra of the dry samples were also presented at the bottom for comparisons. (b) The Raman A_{1g} peaks of Co₃O₄ (top) and Co₃O₄/CeO₂ (bottom) were fitted with Lorentzian function to extract the peak positions, intensity, and FWHM (dash lines: raw spectra; dots: fitting results). (c) The Raman A_{1g} peak positions (upper panel) and intensity ratios with respect to the initial intensity at 1.22 V (lower panel) plotted against the applied potential. The open symbols represent the data collected at 1.22 V at the end after applying the higher potential sequence.

To understand the evolution of the local bonding environment at the catalyst surface during the OER process, the peak position, intensity, and full width at half maximum (FWHM) of the Raman A_{1g} peak (~ 690 cm⁻¹) were extracted by fitting with Lorentzian function (Figure R7b,c). The shift in the peak position as a function of applied potential can be interpreted as either the change in crystalline domain size or the generation of surface strain/stress (i.e. lattice contraction/extension)^{4,5}. Since the marginal variations in the peak FWHM suggest the crystalline domain sizes of both samples remain relatively constant during the OER process (Figure R8, new supplementary Fig. 26), the observed peak position shift should result from the lattice contraction/extension and the surface reconstruction due to the changing local bonding environments. More importantly, the peak positions shift in opposite directions on these two catalysts as the potential goes over the OER catalytic onsets (Figure R7c upper panel). Co₃O₄/CeO₂ shows a red-shift in the A_{1g} peak position after the onset of OER at 1.52 V vs. RHE. Red-shifts in Raman signals are commonly observed in OER catalysts (CoO_x^{3,6}, NiOOH⁷, NiFe and CoFe oxyhydroxides⁸) at OER operating potentials, and they generally reflect the

characteristic vibration for local bonding environment at the outer layer of catalysts with oxidized active site during OER. Thus the generation of active Co^{IV} species that can participate in a fast and efficient OER process should lead to the observed red-shift of the Raman signals. In contrast, blue-shifts in Raman signals usually suggest lattice contraction and charge redistribution^{4,9}. Unlike the more active $\text{Co}_3\text{O}_4/\text{CeO}_2$, the pure Co_3O_4 catalyst would go through a significant charge-accumulation surface reconstruction ($\text{Co}^{\text{III}}\text{Co}^{\text{IV}} \leftrightarrow \text{Co}^{\text{IV}}\text{Co}^{\text{IV}}$) at ~ 1.62 V around the onset for OER. The Co^{IV} species generated during this process are stabilized and cannot participate in fast OER turnover since the reduction peak could be still observed when the potential was scanned backwards, thus they lead to a blue-shift in the Raman signals (Figure R7c). Another interesting difference is that the peak position of $\text{Co}_3\text{O}_4/\text{CeO}_2$ at 1.22 V vs. RHE remains almost unchanged before and after applying the higher potential sequence, suggesting the flexibility in the local bonding environment of Co_3O_4 in the composite catalyst. However, the peak position of Co_3O_4 cannot fully recover after the same potential cycle, with the final peak at ~ 1 cm^{-1} higher in wavenumber (Figure R9, shown on page 14), which is consistent with the positive charge accumulated at the Co center with shorter Co-O bond in the Co_3O_4 sample after OER (Fig. 4a-c). Together with the *ex situ* XAS results, the *in situ* Raman results clearly demonstrate that the bonding environments surrounding the Co centers are affected by the electronic modulation between Co_3O_4 and CeO_2 , which allows the Co active sites to be more readily oxidized and avoid the significant potential-determining surface reconstruction (forming dimeric $\text{Co}^{\text{IV}}\text{Co}^{\text{IV}}$) with charge accumulation and lattice contraction. As Co^{IV} is the key intermediate to start OER process, the more facile formation of Co^{IV} species and de-stabilization of $\text{Co}^{\text{IV}}\text{Co}^{\text{IV}}$ in $\text{Co}_3\text{O}_4/\text{CeO}_2$ would allow faster OER kinetics thus enhance the catalytic activity.

Figure R8. The FWHM of the A_{1g} peaks from the *in situ* Raman spectra for Co_3O_4 and $\text{Co}_3\text{O}_4/\text{CeO}_2$ plotted against the applied potential show relatively small variations. Larger FWHM suggests smaller crystallites in $\text{Co}_3\text{O}_4/\text{CeO}_2$, which is consistent with the XRD and HRTEM results.

The Figure R7 has been added as the new Figure 5 in the revised manuscript and significant discussion about the insights gained from these *in situ* Raman experiments surrounding this new Figure 5 has been added to page 11-13 of the revised manuscript. Other figures about the details of

the *in situ* Raman results are added as new Supplementary Figures 25-27.

II) With regard to “*enhances the intrinsic activity of the same type of catalytic active site in Co₃O₄ by modifying the entropy of activation and the concentration of active sites*”. The entropy of activation reflects the number of active-sites that enter in the rate-determining step overpotential¹⁰⁻¹². This was originally shown in the SI of the original manuscript, now Figure S20. It is generally accepted the formation of Co^{IV} is the key intermediates for OER^{2,13,14}. Now the new *in situ* Raman results (see the discussion above) further suggest the oxidation of Co^{III} to Co^{IV} in Co₃O₄/CeO₂ is more facile compared to pure Co₃O₄ (Figure R7a). More importantly, the significant Co^{III}Co^{IV} ↔ Co^{IV}Co^{IV} redox due to surface reconstruction also suggest part of the Co^{IV} in the pure Co₃O₄ is stabilized and cannot participate in the fast catalytic turnover process. Generally, the Co₃O₄/CeO₂ can have more OER-active Co^{IV} species and larger apparent pre-exponential factor at the same overpotential, suggesting a modification of entropy of activation after the introduction of CeO₂. To make it clearer, we have further revised the explanations in the text in page 9.

III) “*Interface effect*” involving CoCe redox couple. We have shown the CV of CeO₂ in Figure R5a / Supplementary Fig. 9b previously, which confirmed that CeO₂ shows very poor OER activity compared to Co₃O₄. More importantly, there is no obvious redox signal in the CeO₂. It is reasonable that the Ce in CeO₂ should be 4+ and will not be further oxidized. We agree with the Reviewer the potential importance of “interface effect” in the Co₃O₄/CeO₂ composite catalysts. But we believe that what we are discussing here are in fact the “interface effect”: the introduction of many nanocrystalline CeO₂ domains next to the nanocrystalline Co₃O₄ catalysts modify the electronic structures and local bonding structures of Co₃O₄ through the interface they share, it is just that we could not fully describe the atomic/molecular details across these interfaces, partially because we do not have the appropriate techniques to fully characterize such interfaces (keep in mind the <10% percentage of already very small nanocrystalline domains). What we do understand clearly are the changes in the atomic structures of the predominant Co₃O₄ domains and the oxidation states of the Co actives during OER processes due to the presence of such Co₃O₄/CeO₂ interface. And we hypothesize that those changes are due to the electronic modulation due to the introduction of CeO₂ into the Co₃O₄/CeO₂ nanocomposite catalyst. Perhaps some of the issues here are merely a matter of semantics and perspectives. We appreciate the Reviewer for the suggestion and we added or revised some language in page 11 and page 15 (conclusion part) to include a broader perspective.

It is acknowledged that the authors have reported a highly active earth-abundant OER electrocatalyst in acidic medium, with thorough electrochemical characterization. However, the true origin of this effect remains largely unknown. In addition to in situ and/or operando techniques (that also have to cope with the typically minute number of active sites), theoretical studies may be crucial to solve this puzzle.

Response: We thank the Reviewer for the positive comments about the high catalytic activity and

thorough study. Following the suggestions from all Reviewers, we have conducted many additional structural characterizations to more clearly elucidate the microstructural details of the nanocomposite Co₃O₄/CeO₂ catalyst. We have also further performed *in situ* Raman experiments, which are pivotal to build the connections between the changes in bonding environments, surface reconstructions, redox features, and the electronic modulations in the nanocomposite catalyst. The new *in situ* experiments, together with the existing *ex situ* X-ray spectroscopy and other structural characterizations, consistently reveal the microstructural states of the catalysts and their changes through the OER processes. Combining this information with the extensive electrochemical studies about the kinetic factors of the OER reaction processes paints a coherent picture on the origin of the catalytic activity enhancement due to the Co₃O₄/CeO₂ nanocomposite catalyst.

References cited in the Response Letter.

- 1 Xiao, Z. *et al.* Operando Identification of the Dynamic Behavior of Oxygen Vacancy-Rich Co₃O₄ for Oxygen Evolution Reaction. *J. Am. Chem. Soc.* **142**, 12087-12095, (2020).
- 2 Moysiadou, A., Lee, S., Hsu, C.-S., Chen, H. M. & Hu, X. Mechanism of Oxygen Evolution Catalyzed by Cobalt Oxyhydroxide: Cobalt Superoxide Species as a Key Intermediate and Dioxygen Release as a Rate-Determining Step. *J. Am. Chem. Soc.* **142**, 11901-11914, (2020).
- 3 Pasquini, C., D'Amario, L., Zaharieva, I. & Dau, H. Operando Raman spectroscopy tracks oxidation-state changes in an amorphous Co oxide material for electrocatalysis of the oxygen evolution reaction. *J. Chem. Phys.* **152**, 194202, (2020).
- 4 Xu, C. Y., Zhang, P. X. & Yan, L. Blue shift of Raman peak from coated TiO₂ nanoparticles. *J. Raman Spectrosc.* **32**, 862-865, (2001).
- 5 Scamarcio, G., Lugará, M. & Manno, D. Size-dependent lattice contraction in CdS_{1-x}Se_x nanocrystals embedded in glass observed by Raman scattering. *Physical Review B* **45**, 13792-13795, (1992).
- 6 Yeo, B. S. & Bell, A. T. Enhanced Activity of Gold-Supported Cobalt Oxide for the Electrochemical Evolution of Oxygen. *J. Am. Chem. Soc.* **133**, 5587-5593, (2011).
- 7 Garcia, A. C., Touzalin, T., Nieuwland, C., Perini, N. & Koper, M. T. M. Enhancement of Oxygen Evolution Activity of Nickel Oxyhydroxide by Electrolyte Alkali Cations. *Angew. Chem. Int. Ed.* **58**, 12999-13003, (2019).
- 8 Bo, X., Li, Y., Chen, X. & Zhao, C. Operando Raman Spectroscopy Reveals Cr-Induced-Phase Reconstruction of NiFe and CoFe Oxyhydroxides for Enhanced Electrocatalytic Water Oxidation. *Chem. Mater.* **32**, 4303-4311, (2020).
- 9 Iqbal, M. W., Shahzad, K., Akbar, R. & Hussain, G. A review on Raman finger prints of doping and strain effect in TMDCs. *Microelectron. Eng.* **219**, 111152, (2020).
- 10 Anderson, A. B. *et al.* Activation Energies for Oxygen Reduction on Platinum Alloys: Theory and Experiment. *J. Phys. Chem, B* **109**, 1198-1203, (2005).
- 11 Duan, Y. *et al.* Revealing the Impact of Electrolyte Composition for Co-Based Water Oxidation Catalysts by the Study of Reaction Kinetics Parameters. *ACS Catal.* **10**, 4160-4170, (2020).
- 12 Shinagawa, T. & Takanabe, K. New Insight into the Hydrogen Evolution Reaction under Buffered Near-Neutral pH Conditions: Enthalpy and Entropy of Activation. *J. Phys. Chem. C* **120**, 24187-24196, (2016).
- 13 Zhang, M., de Respinis, M. & Frei, H. Time-resolved observations of water oxidation

- intermediates on a cobalt oxide nanoparticle catalyst. *Nat. Chem.* **6**, 362-367, (2014).
- 14 Bergmann, A. *et al.* Unified structural motifs of the catalytically active state of Co(oxyhydr)oxides during the electrochemical oxygen evolution reaction. *Nat. Catal.* **1**, 711-719, (2018).

REVIEWER COMMENTS

Reviewer #1 (Remarks to the Author):

The authors have addressed all questions and concerns in this revised version of their original work. Furthermore, they have properly solved the few problems the reviewers found in their initial conclusions. The result is a solid, highly interesting manuscript that will attract much attention, in my opinion. So, I am happy to support publication in its present form.

Reviewer #2 (Remarks to the Author):

The authors took very carefully into consideration the reports and the critical questions and concerns of the second and third reviewers. The characterization of the pristine samples has been slightly improved by including some additional data and improving the analysis of e.g. the TEM images. Although, the authors could not improve the quality of the EDX maps to resolve single ceria and quite crystallites, the characterization of the pristine samples can now be considered sufficient. To answer the main questions of the two reviewers about the role of ceria nanocrystals, the authors tried to apply operando Raman spectroscopy. Although, the origin of the effect is still not fully clear, even after introducing these additional results, it is also acknowledged that the authors report a relatively stable and highly active oxygen evolution electrocatalyst made of earth-abundant elements, which is active in acidic media. Further studies will be necessary to fully elucidate the role of ceria nanocrystals, but it will involve efforts and techniques, which are definitely not at the reach of the authors at this point.

Reviewer #3 (Remarks to the Author):

The authors have extended the manuscript and resolved the initial confusion due to using "incorporation" alternating with "nanocomposite". Additional in situ Raman measurements were added that again demonstrate the difference between Co_3O_4 and $\text{Co}_3\text{O}_4/\text{CeO}_2$.

Nevertheless, this still does still not directly PROVE an electronic interaction, which is the main claim auf the article and the basis of suggesting a novel strategy (although mixed oxides are very common in thermal catalysis).

Concerning Raman, even the authors themselves state "The shift in the peak position as a function of applied potential can be interpreted as either the change in crystalline domain size or the generation of strain/stress (i.e. lattice contraction/extension)".

"Electronic interactions" would show up in a modified valence band, which could be measured by UV photoemission or would be accessible by DFT modeling. In the current manuscript, such interactions are a plausible hypothesis, but remain speculative.

Furthermore, it is really difficult to imagine that the interfaces between the Co_3O_4 and CeO_2 crystals in the nanocomposite are solely responsible for the reported effect. The interaction is restricted to the tri-phase-boundary and its contribution to the overall surface area of the nanocomposite is very small. Also, there is no chemical bonding across adjoining crystal faces. Based on the leaching experiments, showing pronounced solubility of Co, it seems rather plausible that atomically dispersed Co – that may be incorporated in the ceria surface and that would go undetected by TEM etc. – is in fact responsible for the observed improved OER performance.

The authors still focus on the "electronic modulation", which is not proven at all in the article ("...clearly demonstrate the bonding environment surrounding Co centers are affected by the electronic

modulation between Co₃O₄ and CeO₂ across the interface ..." and "... likely electronic modulation between Co₃O₄ and CeO₂ across the Co₃O₄/CeO₂ interface in the nanocomposite catalyst creates a more favorable local bonding environment in Co₃O₄ that allows the Co^{III} surface species to be easily oxidized into OER-active Co^{IV} species and suppresses the charge accumulation of Co₃O₄ under electrochemical conditions")

Furthermore, in the title and the abstract the term "modulation" appears, but it is not appropriate here and just (mis)used as synonym of "change" or "modification". "Modulation" has a specific physical meaning and this is not a matter of semantics.

Response to Referees

We thank all three Reviewers for their positive reviews and the constructive suggestions that help to improve the scientific presentation of this manuscript. Point-by-Point responses to address the concerns raised by the Reviewers are shown below:

Reviewer #1 (Remarks to the Author)

The authors have addressed all questions and concerns in this revised version of their original work. Furthermore, they have properly solved the few problems the reviewers found in their initial conclusions. The result is a solid, highly interesting manuscript that will attract much attention, in my opinion. So, I am happy to support publication in its present form.

Response: We sincerely thank the Reviewer for the approval of this manuscript.

Reviewer #2 (Remarks to the Author):

The authors took very carefully into consideration the reports and the critical questions and concerns of the second and third reviewers. The characterization of the pristine samples has been slightly improved by including some additional data and improving the analysis of e.g. the TEM images. Although, the authors could not improve the quality of the EDX maps to resolve single ceria and quite crystallites, the characterization of the pristine samples can now be considered sufficient. To answer the main questions of the two reviewers about the role of ceria nanocrystals, the authors tried to apply operando Raman spectroscopy. Although, the origin of the effect is still not fully clear, even after introducing these additional results, it is also acknowledged that the authors report a relatively stable and highly active oxygen evolution electrocatalyst made of earth-abundant elements, which is active in acidic media. Further studies will be necessary to fully elucidate the role of ceria nanocrystals, but it will involve efforts and techniques, which are definitely not at the reach of the authors at this point.

Response: We appreciate the positive comments from the Reviewer. We understand the changes in the local bonding environment observed from the XAS and Raman characterizations cannot be directly linked to the electronic interactions between Co_3O_4 and $\text{Co}_3\text{O}_4/\text{CeO}_2$. To further address this concern, we have collected the ultraviolet photoelectron spectroscopy (UPS) data following the suggestions from *Reviewer #3*, as shown in Figure R1 (also presented as the new Supplementary Figure 23). The related discussions have also been added on page S25 in the Supplementary Information.

“Besides the variations in the intensity of UPS spectra, the differences between Co_3O_4 and $\text{Co}_3\text{O}_4/\text{CeO}_2$ are also observed in the cutoff energy (E_{cutoff}) for the secondary electrons and at the valence band edge (E_{edge} , reflecting the difference between the Fermi level and the valence band

maximum). The work function ($\Phi = h\nu - E_{\text{cutoff}}^{1,2}$, where $h\nu$ is 21.22 eV for the excitation energy of the He I source) of Co_3O_4 and $\text{Co}_3\text{O}_4/\text{CeO}_2$ can be determined to be 4.85 and 4.98 eV, with the corresponding valence band energy [$E_{\text{VB}} = -(\Phi + E_{\text{edge}})$ vs. vacuum^{1,2}] of -5.25 and -5.20 eV, respectively. The work function of CeO_2 was calculated to be 5.287 eV according to the literature³. Thus, the slightly modified work function of $\text{Co}_3\text{O}_4/\text{CeO}_2$ compared to Co_3O_4 can result from the charge redistribution between Co_3O_4 and CeO_2 to reach the equilibrium state. The charge redistribution across the $\text{Co}_3\text{O}_4/\text{CeO}_2$ nanocomposite interface can also be reflected by the higher average Co valence state and shorter Co-O bond distance from XAS results (Figure 4). The similar phenomenon has also been reported by Liu et al previously². These results suggest possible electronic interactions in $\text{Co}_3\text{O}_4/\text{CeO}_2$ nanocomposite.”

Figure R1. The UPS spectra of Co_3O_4 and $\text{Co}_3\text{O}_4/\text{CeO}_2$ on FTO, the inset shows the enlarged spectra near the Fermi edge to highlight the difference at the valence band edge (E_{edge}).

In addition, new comments on the UPS data have also been added on page 10 in the main text.

“Ultraviolet photoelectron spectroscopy (UPS) (Supplementary Fig. 23) showed a larger work function in $\text{Co}_3\text{O}_4/\text{CeO}_2$ than pure Co_3O_4 , suggesting that the electronic structure in $\text{Co}_3\text{O}_4/\text{CeO}_2$ was slightly modified due to possible electronic interactions between Co_3O_4 and CeO_2 ”. We hope the new finding would be helpful to enhance the understanding on the interactions between Co_3O_4 and CeO_2 . Furthermore, in consideration of Reviewer 3’s comments, we have also softened the claims related to the “electronic modulations/interactions” throughout the manuscript. The major revisions can be found in track changes.

Reviewer #3 (Remarks to the Author):

The authors have extended the manuscript and resolved the initial confusion due to using “incorporation” alternating with “nanocomposite”. Additional in situ Raman measurements were added that again demonstrate the difference between Co_3O_4 and $\text{Co}_3\text{O}_4/\text{CeO}_2$. Nevertheless, this still does not directly PROVE an electronic interaction, which is the main claim of the article

and the basis of suggesting a novel strategy (although mixed oxides are very common in thermal catalysis).

Response: We thank the Reviewer for the approval on the revisions of this manuscript, especially our new *in situ* Raman measurements. We agree on the lack of direct evidences to conclusively connect the modified local bonding environments observed from *ex situ* XAS and *in situ* Raman characterizations to the electronic interactions between Co_3O_4 and CeO_2 . As suggested by the Reviewer, we have further collected the ultraviolet photoelectron spectroscopy (UPS) data to further study the potential changes of electronic structure in the $\text{Co}_3\text{O}_4/\text{CeO}_2$ composites, which will be discussed in detail below in the **Response to Comment 2**. Still, this is not *in situ* data. To avoid the over-interpretation on these results, we decided to soften the claims on “electronic modulations/interactions” throughout the manuscript. We thank the constructive suggestions from the Reviewer, we have followed them and further improved the rigor of the discussions in the manuscript. Please find our detailed responses and the major changes made in the Response to each specific comment below:

1. Concerning Raman, even the authors themselves state “The shift in the peak position as a function of applied potential can be interpreted as either the change in crystalline domain size or the generation of strain/stress (i.e. lattice contraction/extension)”.

Response: Thanks for pointing this out. Usually, the Raman peak will be red-shifted with broadening in the FWHM when the crystallinity decreases dramatically. To confirm the shift of the Raman peak is not mostly attributed from the crystalline domain size, we had also extracted the FWHM, as demonstrated in the Supplementary Figure 27. Obviously, the changes in the FWHM are small so the observed shift in peak position is attributed to the generation of strain/stress, which would be reflected by the contraction/extension of the lattice, as also evidenced by the XAS results. To make this point more clear, we have further revised this sentence on page 12 in the main text as “The shift in the peak position as a function of applied potential can be interpreted as either the change in crystallinity (e.g. red-shift with broadening in FWHM happens when the crystallinity decreases dramatically), or the generation of strain/stress (i.e. lattice contraction/extension). Since the marginal variations in the peak FWHM suggested the crystalline domain sizes of both samples remain relatively constant during the OER process (Supplementary Fig. 27), the observed peak position shift should result from the lattice contraction/extension and surface reconstruction due to the changing local bonding environments.”

2. “Electronic interactions” would show up in a modified valence band, which could be measured by UV photoemission or would be accessible by DFT modeling. In the current manuscript, such interactions are a plausible hypothesis, but remain speculative.

Response: Thanks for the great suggestion! We have further collected the UPS data to study the possible electronic interactions in the $\text{Co}_3\text{O}_4/\text{CeO}_2$ composites, as shown in Figure R1 below (also presented as the new Supplementary Figure 23). Slight differences were observed in the UPS spectra, especially the work function of $\text{Co}_3\text{O}_4/\text{CeO}_2$ was modified compared to that of the Co_3O_4 ,

suggesting possible electronic interactions. Accordingly, we have added the following discussions on page S25 in the Supplementary Information.

“Besides the variations in the intensity of UPS spectra, the differences between Co_3O_4 and $\text{Co}_3\text{O}_4/\text{CeO}_2$ are also observed in the cutoff energy (E_{cutoff}) for the secondary electrons and at the valence band edge (E_{edge} , reflecting the difference between the Fermi level and the valence band maximum). The work function ($\Phi = h\nu - E_{\text{cutoff}}$ ^{1,4}, where $h\nu$ is 21.22 eV for the excitation energy of the He I source) of Co_3O_4 and $\text{Co}_3\text{O}_4/\text{CeO}_2$ can be determined to be 4.85 and 4.98 eV, with the corresponding valence band energy [$E_{\text{VB}} = -(\Phi + E_{\text{edge}})$ vs. vacuum^{1,4}] of -5.25 and -5.20 eV, respectively. The work function of CeO_2 was calculated to be 5.287 eV according to the literature³. Thus, the slightly modified work function of $\text{Co}_3\text{O}_4/\text{CeO}_2$ compared to Co_3O_4 could result from the charge redistribution between Co_3O_4 and CeO_2 to reach the equilibrium state. The charge redistribution across the $\text{Co}_3\text{O}_4/\text{CeO}_2$ nanocomposite interface can also be reflected by the higher average Co valence state and shorter Co-O bond distance from the XAS results (Figure 4). Similar phenomenon has also been reported by Liu et al previously². These results suggest possible electronic interaction in $\text{Co}_3\text{O}_4/\text{CeO}_2$ nanocomposite.”

Figure R1. The UPS spectra of Co_3O_4 and $\text{Co}_3\text{O}_4/\text{CeO}_2$, the inset shows the enlarged spectra near the Fermi edge to highlight the difference at the valence band edge (E_{edge}).

In addition, new comments on the UPS data have also been added on page 10 in the main text.

“Ultraviolet photoelectron spectroscopy (UPS) (Supplementary Fig. 23) showed a larger work function in $\text{Co}_3\text{O}_4/\text{CeO}_2$ than pure Co_3O_4 , suggesting that the electronic structure in $\text{Co}_3\text{O}_4/\text{CeO}_2$ was slightly modified due to possible electronic interactions between Co_3O_4 and CeO_2 ”.

We also agree that we could not yet conclusively prove the electronic interactions between Co_3O_4 and CeO_2 under OER operation conditions, thus we have softened the claims related to “electronic modulations/interactions” throughout the manuscript, which will be discussed in detail in the *Response to Comment 4*.

3. Furthermore, it is really difficult to imagine that the interfaces between the Co_3O_4 and CeO_2 crystals in the nanocomposite are solely responsible for the reported effect. The interaction is restricted to the tri-phase-boundary and its contribution to the overall surface area of the nanocomposite is very small. Also, there is no chemical bonding across adjoining crystal faces. Based on the leaching experiments, showing pronounced solubility of Co, it seems rather plausible that atomically dispersed Co – that may be incorporated in the ceria surface and that would go undetected by TEM etc. – is in fact responsible for the observed improved OER performance.

Response: We thank the Reviewer for providing this alternative explanation for the enhanced performance in the $\text{Co}_3\text{O}_4/\text{CeO}_2$. After carefully checking all the results we presented, we believe that this proposed model could not fit the observations presented here. We have proved that Co_3O_4 and $\text{Co}_3\text{O}_4/\text{CeO}_2$ have similar active sites by analysis on apparent activation energy (Figure 3f). They also showed comparable stability from both electrocatalytic performance and Co leaching behaviors. Combining with *in situ* Raman results, we have coherently proved the enhanced acidic OER performance in $\text{Co}_3\text{O}_4/\text{CeO}_2$ was due to the fact that the Co^{III} species are easier to be oxidized into OER-active Co^{IV} species. We do not think we could deduce that the Co is atomically dispersed on the ceria surface simply according to the leaching experiment. It has been proven that ceria was highly stable at the open circuit potential (Supplementary Fig. 31a) but was not as stable at the OER working potential (Supplementary Fig. 30b). As suggested by the Reviewer during the first round revision and also evidenced in Supplementary Table 1 and Supplementary Figure 30b, the CeO_2 at the surface were more easily dissolved. In that case, the atomically dispersed Co on ceria would lose the active sites very quickly and should not have comparable stability. During the first round revision, we have also proved that the proposed model of Ce atom dispersed on the Co_3O_4 is not the case for our catalyst. Thus, we hope that we have definitely cleared out this uncertainty about the catalyst model for $\text{Co}_3\text{O}_4/\text{CeO}_2$, which is a composite in nature.

We also want to emphasize that the Co_3O_4 (about 9.4 nm) and CeO_2 (about 5 nm) domains are very small and highly mixed (Supplementary Fig. 4), which would significantly increase the interfacial contact area and enlarge the impact of the interactions. Even though we could confirm rearranged chemical bonding at the interface, the charge rearrangement (e.g. caused by the difference in the work function) across the interface will significantly change the structural properties, as verified by the surface-sensitive techniques such as Raman (Figure 5), XPS (Supplementary Fig. 22) and UPS (Supplementary Fig. 23), as well as the bulk-sensitive XAS technique (Figure 4). Due to limitations from the characterization techniques, we cannot quantitatively identify the maximum thickness that such interfacial interactions will affect in the $\text{Co}_3\text{O}_4/\text{CeO}_2$ nanocomposites. But we hope these explanations based on all these characterization results are helpful to strengthen the understanding on interfacial impacts in $\text{Co}_3\text{O}_4/\text{CeO}_2$.

4. The authors still focus on the “electronic modulation”, which is not proven at all in the article (“...clearly demonstrate the bonding environment surrounding Co centers are affected by the electronic modulation between Co_3O_4 and CeO_2 across the interface ...” and “... Likely electronic modulation between Co_3O_4 and CeO_2 across the $\text{Co}_3\text{O}_4/\text{CeO}_2$ interface in the nanocomposite

catalyst creates a more favorable local bonding environment in Co_3O_4 that allows the Co^{III} surface species to be easily oxidized into OER-active Co^{IV} species and suppresses the charge accumulation of Co_3O_4 under electrochemical conditions”). Furthermore, in the title and the abstract the term “modulation” appears, but it is not appropriate here and just (mis)used as synonym of “change” or “modification”. “Modulation” has a specific physical meaning and this is not a matter of semantics.

Response: We thank the Reviewer for bringing this to our attention! We changed the descriptions and softened our claims, since the word “modulation” could unnecessarily mislead readers from different research backgrounds. We appreciate the constructive suggestions that significantly improve the rigor of this manuscript. As discussed above in the *Response to Comment 2*, we have collected the UPS data to probe possible electronic interactions. But we also decided to soften the claims on “electronic modulation/interaction” throughout the manuscript due to the lack of conclusive *in situ* experimental data that directly links different results. The major revisions include:

1. The title “*Modulating redox properties and local bonding of Co_3O_4 by CeO_2 enhances oxygen evolution catalysis in acid*” was changed to “**Modifying redox properties and local bonding of Co_3O_4 by CeO_2 enhances oxygen evolution catalysis in acid**”.

2. In the abstract, “*The local bonding environment of Co_3O_4 can be modified likely due to the electronic modulation between Co_3O_4 and CeO_2 , which allows the Co^{III} species to be easily oxidized into OER-active Co^{IV} species...*” was changed to “**The local bonding environment of Co_3O_4 can be modified after the introduction of nanocrystalline CeO_2 , which allows the Co^{III} species to be easily oxidized into OER-active Co^{IV} species...**”.

3. In the conclusion, “*...The likely electronic modulation between Co_3O_4 and CeO_2 across the $\text{Co}_3\text{O}_4/\text{CeO}_2$ interface in the nanocomposite catalyst creates a more favorable local bonding environment in Co_3O_4 ...*” was changed to “**...The introduction of nanocrystalline CeO_2 modifies the electronic structures and creates a more favorable local bonding environment in Co_3O_4 ...**”.

For other revisions throughout the text, please also check the track changes. We sincerely thank the Reviewer for your help to improve the scientific presentation of this manuscript.

References cited in the Response Letter

- 1 Liu, F. *et al.* Direct Z-Scheme Hetero-phase Junction of Black/Red Phosphorus for Photocatalytic Water Splitting. *Angew. Chem. Int. Ed.* **58**, 11791-11795, (2019).
- 2 Liu, Y. *et al.* 2D Electron Gas and Oxygen Vacancy Induced High Oxygen Evolution Performances for Advanced $\text{Co}_3\text{O}_4/\text{CeO}_2$ Nanohybrids. *Adv. Mater.* **31**, 1900062, (2019).
- 3 Goldsby, J. Basic Elastic Properties Predictions of Cubic Cerium Oxide Using First-Principles Methods. *Journal of Ceramics* **2013**, 1-4, (2013).
- 4 Liu, W. *et al.* Single-Site Active Cobalt-Based Photocatalyst with a Long Carrier Lifetime for

Spontaneous Overall Water Splitting. *Angew. Chem. Int. Ed.* **56**, 9312-9317, (2017).

REVIEWERS' COMMENTS

Reviewer #3 (Remarks to the Author):

I congratulate the authors on these revisions ! Really well-done. Of course, not everything can be done in situ, but the UPS just gives a very useful hint in the right direction, even if the difference is small. I also appreciate that the "modulation" disappeared. I know it is often (mis)used by others, but I think now your paper is much more solid. Best wishes for the future !